# View-Independent 3D Feature Distillation with Object-Centric Priors

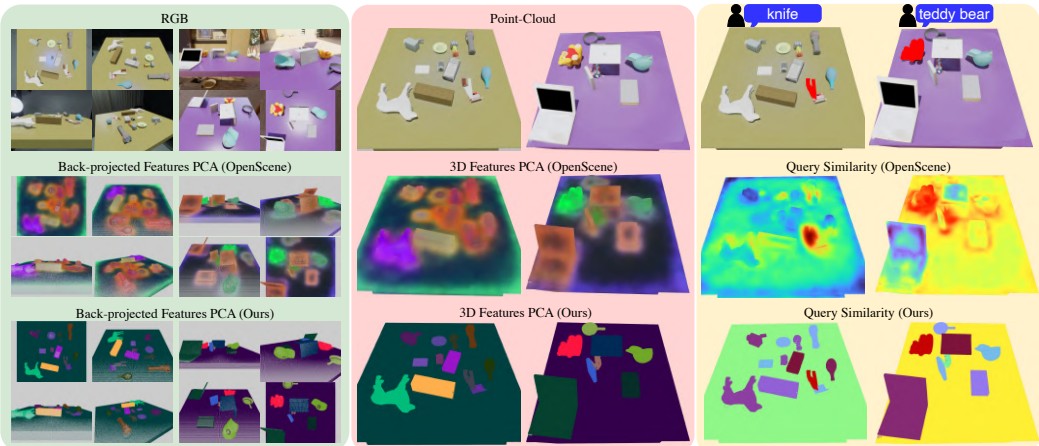

**Figure 1:** Visualization of 3D features *(middle)*, back-projected 2D features *(left)* and user query similarity heatmaps *(right)*, for previous SOTA point-cloud feature distillation method OpenScene and our DROP-CLIP. OpenScene fuses pixel-wise 2D features across all views with average pooling, leading to grounding failures, segmentation imprecisions and fuzzy object boundaries. Our method resolves these issues by employing object-centric priors to fuse object-level 2D features in 3D instance masks with semantics-informed view selection.

## Abstract

Grounding natural language to the physical world is a ubiquitous topic with a wide range of applications in computer vision and robotics. Recently, 2D vision-language models such as CLIP have been widely popularized, due to their impressive capabilities for open-vocabulary grounding in 2D images. Subsequent works aim to elevate 2D CLIP features to 3D via feature distillation, but either learn neural fields that are scene-specific and hence lack generalization, or focus on indoor room scan data that require access to multiple camera views, which is not practical in robot manipulation scenarios. Additionally, related methods typically fuse features at pixel-level and assume that all camera views are equally informative. In this work, we show that this approach leads to sub-optimal 3D features, both in terms of grounding accuracy, as well as segmentation crispness. To alleviate this, we propose a multi-view feature fusion strategy that employs object-centric priors to eliminate uninformative views based on semantic information, and fuse features at object-level via instance segmentation masks. To distill our object-centric 3D features, we generate a large-scale synthetic multi-view dataset of cluttered tabletop scenes, spawning 15k scenes from over 3300 unique object instances, which we make publicly available. We show that our method reconstructs 3D CLIP features with improved grounding capacity and spatial consistency, while doing so from single-view RGB-D, thus departing from the assumption of multiple camera views at test time. Finally, we show that our approach can generalize to novel tabletop domains and be re-purposed for 3D instance segmentation without fine-tuning, and demonstrate its utility for language-guided robotic grasping in clutter.

## 1 Introduction

Language grounding in 3D environments plays a crucial role in realizing intelligent systems that can interact naturally with the physical world. In the robotics field, being able to precisely segment

desired objects in 3D based on open language queries (object semantics, visual attributes, affordances, etc.) can serve as a powerful proxy for enabling open-ended robot manipulation. As a result, research focus on 3D segmentation methods has seen growth in recent years (Chen et al., 2020; Achlioptas et al., 2020b; Luo et al., 2022; Huang et al., 2021; Qian et al., 2024; Takmaz et al., 2023). However, related methods fall in the closed-vocabulary regime, where only a fixed list of classes can be used as queries. Inspired by the success of open-vocabulary 2D methods (Radford et al., 2021; Dong et al., 2022; Ghiasi et al., 2021; Li et al., 2022a), recent efforts elevate 2D representations from pretrained image models (Radford et al., 2021; Oquab et al., 2023) to 3D via distillation pipelines (Peng et al., 2022; Kerr et al., 2023; Nguyen et al., 2023; Koch et al., 2024; Shen et al., 2023; Tschernezki et al., 2022; Kobayashi et al., 2022; Engelmann et al., 2024). In this work, we identify several limitations of existing distillation approaches. On the one hand, field-based methods (Kerr et al., 2023; Rashid et al., 2023; Shen et al., 2023; Tschernezki et al., 2022; Kobayashi et al., 2022) offer continuous 3D feature fields, but require to be trained online in specific scenes and hence cannot generalize to novel object instances and compositions, they require a few minutes to train, and need to collect multiple camera views before training, all of which hinder their real-time applicability. On the other hand, original 3D feature distillation methods and follow up work (Peng et al., 2022; Nguyen et al., 2023; Zhang et al., 2023) use room scan datasets (Dai et al., 2017; Ramakrishnan et al., 2021) to distill 2D features fused from multiple views with point-cloud encoders. The distilled features maintain the open-set generalizability of the pretrained model, therefore granting such methods applicable in novel scenes with open vocabularies. However, such approaches assume that 2D features from all views are equally informative, which is not the case in natural indoors scenes, where due to partial visibility and clutter, certain views will lead to noisy representations. 2D features are also typically fused point-wise from ViT patches (Ghiasi et al., 2021; Li et al., 2022a; Dong et al., 2022) or multi-scale crops (Kerr et al., 2023; Takmaz et al., 2023), therefore leading to the so called "patchyness" issue (Qin et al., 2024) (see Fig. 1), where features computed in patches / crops that involve multiple objects lead to fuzzy segmentation boundaries. The latter issue is especially impactful in robot manipulation, where precise 3D segmentation is vital for specifying robust actuation goals.

To address such limitations, in this work, we revisit 2D $\rightarrow$ 3D feature distillation with point-cloud encoders, but revise the multi-view feature fusion strategy to enhance the quality of the target 3D features. In particular, we inject both *semantic* and *spatial* object-centric priors into the fusion strategy, in three ways: (i) We obtain object-level 2D features by isolating object instances in each camera view from their 2D segmentation masks, (ii) we fuse features only at corresponding 3D object regions using their 3D segmentation masks, (iii) we leverage dense object-level semantic information to devise an informativeness metric, which is used to weight the contribution of views and eliminate uninformative ones. Extensive ablation studies demonstrate the advantages of our proposed object-centric fusion strategy compared to vanilla approaches. To train our method, we require a large-scale cluttered indoors dataset with dense number of views per scene, which is currently not existent. To that end, we build **MV-TOD** (**M**ulti-**V**iew **T**abletop **O**bjects **D**ataset), consisting of $\sim 15k$ Blender scenes from more than $3.3k$ unique 3D object models, for which we provide $73$ views per scene with $360°$ coverage, further equipped with 2D/3D segmentations, 6-DoF grasps and semantic object-level annotations. We use MV-TOD to distill the object-centric 3D CLIP (Radford et al., 2021) features acquired via our fusion strategy into a 3D representation, which we call **DROP-CLIP** (**D**istilled **R**epresentations with **O**bject-centric **P**riors from CLIP). Our 3D encoder operates in partial point-clouds from a single RGB-D view, thus departing from the requirement of multiple camera images at test time, while offering real-time inference capabilities. By imposing the same 3D features as distillation targets for a large number of diverse views, we encourage DROP-CLIP to learn a view-invariant 3D representation. We demonstrate that our learned 3D features surpass previous 3D open-vocabulary approaches in semantic and referring segmentation tasks in MV-TOD, both in terms of grounding accuracy and segmentation crispness, while significantly outperforming previous 2D approaches in the single-view setting. Further, we show that they can be leveraged zero-shot in novel tabletop datasets that contain real-world scenes with unseen objects and new vocabulary, as well as be used out-of-the-box for 3D instance segmentation tasks, performing competitively with established segmentation approaches without fine-tuning.

In summary, our contributions are fourfold: (i) we release MV-TOD, a large-scale synthetic dataset of household objects in cluttered tabletop scenarios, featuring dense multi-view coverage and semantic/mask/grasp annotations, (ii) we identify limitations of current multi-view feature fusion approaches and illustrate how to overcome them by leveraging object-centric priors, (iii) we release DROP-CLIP, a 3D model that reconstructs view-independent 3D CLIP features from single-view,

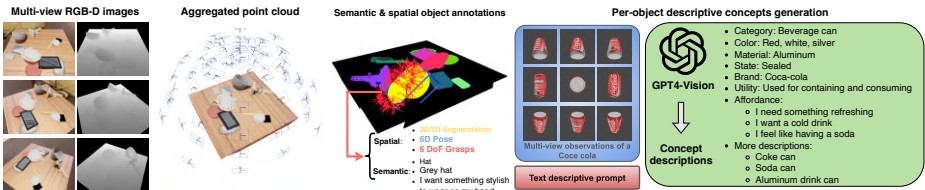

Figure 2: **MV-TOD Overview:** Example generated scene, source multi-view RGB=D images and scene annotations *(left)*. Automatic semantic annotation generation with VLMS *(right)*.

and (iv) we conduct extensive ablation studies, comparative experiments and robot demonstrations to showcase the effectiveness of the proposed method in terms of 3D segmentation performance, generalization to novel domains and tasks, and applicability in robot manipulation scenarios.

## 2 MULTI-VIEW TABLETOP OBJECTS DATASET

Existing 3D datasets mainly focus on indoor scenes in room layouts (Armeni et al., 2016; Dai et al., 2017; Straub et al., 2019) and related annotations typically cover closed-set object categories (e.g. furniture) (Chen et al., 2020; Achlioptas et al., 2020b; Liu et al., 2021; Rozenberszki et al., 2022b; Mauceri

| Dataset | Layout | Multi View | Clutter | Vision Data | Ref.Expr. Annot. | Grasp Annot. | Num.Obj. Categories | Num. Scenes | Num. Expr. | Obj.-lvl Semantics |
|---|---|---|---|---|---|---|---|---|---|---|
| ScanNet (Dai et al., 2017) | indoor | ✔ | - | RGB-D,3D | ✗ | ✗ | 17 | 800 | – | ✗ |
| S3DIS (Chen et al., 2022) | indoor | ✔ | - | RGB-D,3D | ✗ | ✗ | 13 | 6 | – | ✗ |
| Replica (Straub et al., 2019) | indoor | ✔ | - | RGB-D,3D | ✗ | ✗ | 88 | – | – | ✔ |
| STPLS3D (Chen et al., 2022) | outdoor | ✔ | - | 3D | ✗ | ✗ | 12 | 18 | – | ✗ |
| ScanRefer (Chen et al., 2020) | indoor | ✔ | ✗ | RGB-D,3D | 2D/3D mask | ✗ | 18 | 800 | 51.5k | ✗ |
| ReferIt-3D (Achlioptas et al., 2020b) | indoor | ✔ | ✗ | RGB-D,3D | 2D/3D mask | ✗ | 18 | 707 | 125.5k | ✗ |
| ReferIt-RGBD (Liu et al., 2021) | indoor | ✗ | ✗ | RGB-D | 2D box | ✗ | - | 7.6k | 38.4k | ✗ |
| SunSpot (Mauceri et al., 2019) | indoor | ✗ | ✔ | RGB-D | 2D box | ✗ | 38 | 1.9k | 7.0k | ✗ |
| GraspNet (Fang et al., 2020) | tabletop | ✗ | ✔ | 3D | ✗ | 6-DoF | 88 | 190 | – | ✗ |
| REGRAD (Zhang et al., 2022) | tabletop | ✔ | ✔ | RGB-D,3D | ✗ | 6-DoF | 55 | 47k | – | ✗ |
| OCID-VLG (Tziafas et al., 2023) | tabletop | ✗ | ✔ | RGB-D,3D | 2D mask | 4-DoF | 31 | 1.7k | 89.6k | template |
| Grasp-Anything (Vuong et al., 2023) | tabletop | ✗ | ✗ | RGB | 2D mask | 4-DoF | 236 | 1M | – | open |
| MV-TOD (ours) | tabletop | ✔ | ✔ | RGB-D,3D | 3D mask | 6-DoF | 149 | 15k | 671.2k | open |

Table 1: Comparisons between MV-TOD and existing datasets.

et al., 2019), which are not practical for robot manipulation tasks, where cluttered tabletop scenarios and open-vocabulary language are of key importance. Alternatively, recent grasp-related research efforts collect cluttered tabletop scenes, but either lack language annotations (Zhang et al., 2022; Eppner et al., 2020; Fang et al., 2020) or connect cluttered scenes with language but only for 4-DoF grasps with RGB data (Tziafas et al., 2023; Vuong et al., 2023), hence lacking crucial 3D information. Further, all existing datasets lack dense multi-view scene coverage, granting them non applicable for 2D → 3D feature distillation, where we require multiple images from each scene to extract 2D features with a foundation model. To cover this gap, we propose MV-TOD, a large-scale synthetic dataset with cluttered tabletop scenes featuring dense multi-view coverage, segmentation masks, 6-DoF grasps and rich language annotations at the object level (see Fig. 2). Table 1 summarizes key differences between MV-TOD and existing grounding / grasping datasets.

MV-TOD contains approximately $15k$ scenes generated in Blender (Community, 2018), comprising of 3379 unique object models, 99 collected by us and the rest filtered from ShapeNet-Sem model set (Chang et al., 2015). The dataset enumerates 149 object categories featuring typical household objects (kitchenware, food, electronics etc.), each of which includes multiple instances that vary in fine-grained details such as color, texture, shape etc. For each object instance, we leverage modern vision-language models such as GPT-4-Vision (GPT, 2023) to generate textual annotations referring to various object attributes, including category, color, material, state, utility, brand, etc., spawning over $670k$ unique referring instance queries. We refer the reader to Appendix A.1 for details on object statistics and scene generation implementation. For each scene, we provide 73 uniformly distributed views, 2D / 3D instance segmentation masks, 6D object poses, as well as a set of referring expressions sampled from the object-level semantic annotations. Additionally, we provide collision-free 6-DoF grasp poses for each scene object, originating from the ACRONYM dataset (Eppner et al., 2020). In this paper, we leverage the dense multi-view coverage of MV-TOD for 2D → 3D feature distillation. However, given the breadth of labels in MV-TOD, we believe it can serve as a resource for several 3D vision and robotics downstream tasks, including instance segmentation, 6D pose estimation and 6-DoF grasp synthesis. To the best of our knowledge, MV-TOD is the first dataset to combine 3D cluttered scenes with multi-view images, open-vocabulary language and 6-DoF grasp annotations.

Figure 3: **Method Overview:** Given a 3D scene and multiple camera views, we employ three object-centric priors *(in red)* for multi-view feature fusion: (i) extract CLIP features from 2D masked object crops, (ii) use semantic annotations to fuse 2D features across views, (iii) apply the fused feature on all points in the object's 3D mask. The fused feature-cloud is distilled with a single-view posed RGB-D encoder and cosine distance loss. During inference, we compute point-wise cosine similarity scores in CLIP space (higher similarity towards red).

# 3 DISTILLED REPRESENTATIONS WITH OBJECT-CENTRIC PRIORS

Our goal is to distill multi-view 2D CLIP features into a 3D representation, while employing an object-centric feature fusion strategy to ensure high quality 3D features. Our overall pipeline is illustrated in Fig 3. We first introduce traditional multi-view feature fusion (Sec. 3.1), present our variant with object-centric priors (Sec. 3.2), discuss feature distillation training (Sec. 3.3) and describe how to perform inference for downstream open-vocabulary 3D grounding tasks (Sec. 3.4).

## 3.1 MULTI-VIEW FEATURE FUSION

We assume access to a dataset of 3D scenes, where each scene is represented through a set of $\mathcal{V}$ posed RGB-D views $\left\{I_v \in \mathbb{R}^{H \times W \times 3}, D_v \in \mathbb{R}^{H \times W}, T_v \in \mathbb{R}^{4 \times 4}\right\}_{v=1}^{\mathcal{V}}$, with $H \times W$ denoting the image resolution, $\mathcal{V}$ the total number of views, and $T_v$ the transformation matrix from each camera's viewpoint $v$ with respect to a global reference frame, such as the center of the tabletop. A projection matrix $K_v$ representing each camera's intrinsic parameters is also given. For each scene we reconstruct the full point-cloud $P \in \mathbb{R}^{M \times 3}$ by aggregating all depth images $D_v$, after projecting them to 3D with the camera intrinsics $K_v$ and transforming to world frame with $T_v^{-1}$. To remove redundant points, we voxelize the aggregated point-cloud with a fixed voxel size resolution $d^3$, resulting in $M$ total points. Our goal is to obtain a feature-cloud $Z^{3D} \in \mathbb{R}^{M \times C}$, where $C$ is the dimension of the representations provided by the pretrained image model, fused across all views.

**2D feature extraction** We pass each RGB view to a pretrained image model $f^{2D} : \mathbb{R}^{H \times W \times 3} \to \mathbb{R}^{H \times W \times C}$ to obtain pixel-level features $Z_v^{2D} = f^{2D}(I_v)$. Any ViT-based vision foundation model (e.g. DINO-v2 (Oquab et al., 2023)) can be chosen, but we focus on CLIP (Radford et al., 2021), since we want our 3D representation to be co-embedded with language, as to enable open-vocabulary grounding. However, vanilla CLIP features are restrained to image-level, whereas we require dense pixel-level features to perform multi-view fusion. To obtain pixel-wise features, previous works explore fine-tuned CLIP models (Peng et al., 2022; Koch et al., 2024) such as OpenSeg (Ghiasi et al., 2021) or LSeg (Li et al., 2022a), multi-scale crops from anchored points in the image frame (Kerr et al., 2023; Takmaz et al., 2023; Zhang et al., 2023) or MaskCLIP (Dong et al., 2022; Shen et al., 2023), which provides patch-level text-aligned features from CLIP's ViT encoder without additional training. All approaches are compatible with our framework (ablations in Sec. 4.1).

**2D-3D correspondence** Given the $i$-th point in $P$, $\mathbf{x}_i = (x, y, z)$, $i = 1, \dots, M$, we first back-project to each camera view $v$ using: $\tilde{\mathbf{u}}_{v,i} \doteq \mathcal{M}_v(\mathbf{x}_i) = K_v \cdot T_v \cdot \tilde{\mathbf{x}}_i$, where $\tilde{\mathbf{u}} = (u_x, u_y, u_z)^T$ and $\tilde{\mathbf{x}} = (x, y, z, 1)^T$ homogeneous coordinates in 2D camera frame and 3D world frame respectively, and $\mathbf{u} = (u_x, u_y)^T$. The 2D feature for each back-projected point $\mathbf{z}_{v,i}^{2D} \in \mathbb{R}^C$ is then given by:

$$\mathbf{z}_{v,i}^{2D} = f^{2D}\left(I_v(\mathbf{u}_{v,i})\right) = f^{2D}\left(I_v(\mathcal{M}_v(\mathbf{x}_i))\right) \tag{1}$$

For each view, we eliminate points that fall outside of a camera view's FOV by considering only the pixels: $\left\{\tilde{\mathbf{u}}_v = (u_x, u_y, u_z)^T \in \mathcal{M}_v(P) \mid u_z \neq 0, \ u_x/u_z \in [0, W), \ u_y/u_z \in [0, H)\right\}$. It is further

important to maintain only points that are visible from each camera view, as a point might lie within the camera's FOV but in practise be occluded by a foreground object. To eliminate such points, we follow (Peng et al., 2022; Takmaz et al., 2023) and compare the back-projected z coordinate $u_z$ with the sensor depth reading $D_v(u_x, u_y)$. We maintain only points that satisfy: $|u_z - D_v(u_x, u_y)| \leq c_{thr}$, where $c_{thr}$ a fixed hyper-parameter. We compose the FOV and occlusion filtering to obtain a *visibility map* $\Lambda_{v,i} \in \{0, 1\}^{\mathcal{V} \times M}$, which determines whether point $i$ is visible from view $v$.

**Fusing point-wise features** Obtaining a 3D feature for each point $i = 1, \ldots, M$ is achieved by fusing back-projected 2D features $Z_v^{2D}$ with weighted-average pooling:

$$\mathbf{z}_i^{3D} = \frac{\sum_{v=1}^{\mathcal{V}} \mathbf{z}_{v,i}^{2D} \cdot \omega_{v,i}}{\sum_{v=1}^{\mathcal{V}} \omega_{v,i}} \tag{2}$$

where $\omega_{v,i} \in \mathbb{R}$ a scalar weight that represents the importance of view $v$ for point $i$. In practise, previous works consider $\omega_{v,i} = \Lambda_{v,i}$ (Peng et al., 2022), a binary weight for the visibility of each point. In essence, this method assumes that all views are equally informative for each point, as long as the point is visible from that view.

We suggest that naively average pooling 2D features for each point leads to sub-optimal 3D features, as noisy, uninformative views contribute equally, therefore "polluting" the overall representation. In our work we propose to decompose $\omega_{v,i} = \Lambda_{v,i} \cdot G_{v,i}$, where $G_{v,i} \in \mathbb{R}^{\mathcal{V} \times M}$ an informativeness weight that measures the importance of each view for each point. In the next subsection, we describe how to use text data to dynamically compute an informativeness weight for each view based on *semantic* object-level information, as well as how to perform object-wise instead of point-wise fusion.

## 3.2 EMPLOYING OBJECT-CENTRIC PRIORS

Let $\left\{ S_v^{2D} \in \{0, 1\}^{N \times H \times W} \right\}_{v=1}^{\mathcal{V}}$ be view-aligned 2D instance-wise segmentation masks for each scene, where $N$ the total number of scene objects, provided from the training dataset. We aggregate the 2D masks to obtain $S^{3D} \in \{1, \ldots, N\}^M$, such that for each point $i$ we can retrieve the corresponding object instance $n_i = S_i^{3D}$.

**Semantic informativeness metric** Let $\mathcal{Q} = \{Q_k\}_{k=1}^{\mathcal{K}}$, $Q_k \in \mathbb{R}^{N_k \times C}$ be a set of object-specific textual prompts, where $\mathcal{K}$ the number of dataset object instances and $N_k$ the number of prompts for object $k$. We use CLIP's text encoder to embed the textual prompts in $\mathbb{R}^C$ and average them to obtain an object-specific prompt $\mathbf{q}_k = 1/N_k \cdot \sum_{j=1}^{N_k} Q_{k,j}$. For each scene, we map each object instance $n \in [1, N]$ to its positive prompt $\mathbf{q}_n^+$, as well as a set $Q_n^- \doteq \mathcal{Q} - \{\mathbf{q}_n^+\}$ of negative prompts corresponding to all other instances. We define our *semantic informativeness metric* as:

$$G_{v,i} = \cos(\mathbf{z}_{v,i}^{2D}, \mathbf{q}_{n_i}^+) - \max_{\mathbf{q} \sim Q_{n_i}^-} \cos(\mathbf{z}_{v,i}^{2D}, \mathbf{q}) \tag{3}$$

Intuitively, we want a 2D feature from view $v$ to contribute to the overall 3D feature of point $i$ according to how much its similarity with the correct object instance is higher than the maximum similarity to any of the negative object instances, hence offering a proxy for semantic informativeness. We clip this weight to 0 to eliminate views that don't satisfy the condition $G_{v,i} \geq 0$. Plugging in our metric in equation (2) already provides improvements over vanilla average pooling (see Sec. 4.1), however, does not deal with 3D spatial consistency, for which we employ our spatial priors below.

**Object-level 2D CLIP features** For obtaining object-level 2D CLIP features, we isolate the pixels for each object $n$ from each view $v$ from $S_{v,n}^{2D}$ and crop a bounding box around the mask from $I_v$: $\mathbf{z}_{v,n}^{2D} = f_{cls}^{2D}\left(\texttt{cropmask}(I_v, S_{v,n}^{2D})\right)$ (see Appendix A.3 for ablations in CLIP visual prompts). Here we use $f_{cls}^{2D} : \mathbb{R}^{h_n \times w_n \times 3} \to \mathbb{R}^C$, i.e., only the $\texttt{[CLS]}$ feature of CLIP's ViT encoder, to represent an object crop of size $h_n \times w_n$. We can now define our metric from equation (3) also at object-level:

$$G_{v,n} = \cos(\mathbf{z}_{v,n}^{2D}, \mathbf{q}_n^+) - \max_{\mathbf{q} \sim Q_n^-} \cos(\mathbf{z}_{v,n}^{2D}, \mathbf{q}) \tag{4}$$

where $G_{v,n} \in \mathbb{R}^{\mathcal{V} \times N}$ now represents the semantic informativeness of view $v$ for object instance $n$.

**Fusing object-wise features** A 3D object-level feature can be obtained by fusing 2D object-level features across views similar to equation (2):

$$\mathbf{z}_n^{3D} = \frac{\sum_{v=1}^{\mathcal{V}} \mathbf{z}_{v,n}^{2D} \cdot \omega_{v,n}}{\sum_{v=1}^{\mathcal{V}} \omega_{v,n}} = \frac{\sum_{v=1}^{\mathcal{V}} \mathbf{z}_{v,n}^{2D} \cdot \Lambda_{v,n} \cdot G_{v,n}}{\sum_{v=1}^{\mathcal{V}} \Lambda_{v,n} \cdot G_{v,n}} \tag{5}$$

where each view is weighted by its semantic informativeness metric $G_{v,n}$, as well as optionally a visibility metric $\Lambda_{v,n} = \sum S_{v,n}^{2D}$ that measures the number of pixels from $n$-th object's mask that are visible from view $v$ (Takmaz et al., 2023). We finally reconstruct the full feature-cloud $Z^{3D} \in \mathbb{R}^{M \times C}$ by equating each point's feature to its corresponding 3D object-level one via: $\mathbf{z}_i^{3D} = \mathbf{z}_{n_i}^{3D}$, $n_i = S_i^{3D}$.

### 3.3 View-Independent Feature Distillation

Even though the above feature-cloud $Z^{3D}$ could be directly used for open-vocabulary grounding in 3D, its construction is computationally intensive and requires a lot of expensive resources, such as access to multiple camera views, view-aligned 2D instance segmentation masks, as well as textual prompts to compute informativeness metrics. Such utilities are rarely available in open-ended scenarios, especially in robotic applications, where usually only single-view RGB-D images from sensors mounted on the robot are provided. To tackle this, we wish to distill all the above knowledge from the feature-cloud $Z^{3D}$ with an encoder network that receives only a partial point-cloud from single-view posed RGB-D. Hence, the only assumption that we make during inference is access to camera intrinsic and extrinsic parameters, which is a mild requirement in most robotic pipelines.

In particular, given a partial colored point-cloud from view $v$: $P_v \in \mathbb{R}^{M_v \times 6}$, we train an encoder $\mathcal{E}_\theta : \mathbb{R}^{M_v \times 6} \to \mathbb{R}^{M_v \times C}$ such that $\mathcal{E}_\theta(P_v) = Z^{3D}$. Notice that the distillation target $Z^{3D}$ is independent of view $v$. Following (Peng et al., 2022; Koch et al., 2024) we use cosine distance loss:

$$\mathcal{L}(\theta) = 1 - \cos(\mathcal{E}_\theta(P_v), Z^{3D}) \tag{6}$$

See Appendix A.2 for training implementation details. With such a setup, we can obtain 3D features that: (i) are co-embedded in CLIP text space, so they can be leveraged for 3D segmentation tasks from open-vocabulary queries, (ii) are ensured to be optimally informative per object, due to the usage of the semantic informativeness metric to compute $Z^{3D}$, (iii) maintain 3D spatial consistency in object boundaries, due to performing object-wise instead of point-wise fusion when computing $Z^{3D}$, and (iv) are encouraged to be view-independent, as the same features $Z^{3D}$ are utilized as distillation targets regardless of the input view $v$. Importantly, no labels, prompts, or segmentation masks are needed at test-time to reproduce the fused feature-cloud, while obtaining it amounts to a single forward pass of our 3D encoder, hence offering real-time performance.

### 3.4 Open-Vocabulary 3D Segmentation

Given a predicted feature-cloud $\hat{Z}^{3D} = \mathcal{E}_\theta(P_v)$, we can perform 3D grounding tasks from open-vocabularies by computing cosine similarities between CLIP text embeddings and $\hat{Z}^{3D}$.

**Semantic segmentation** In this task, the queries correspond to an open-set of textual prompts $Q = \{\mathbf{q}_k\}_{k=1}^{\mathcal{K}}$ describing $\mathcal{K}$ semantic classes. A class for each point $\hat{Y} \in \{1, \ldots, \mathcal{K}\}^M$ is given by : $\hat{Y} = \text{argmax}_k \cos(\hat{Z}^{3D}, \mathbf{q}_k)$.

**Referring segmentation** Here the user provides an open-vocabulary query $\mathbf{q}^+$ referring to a particular object instance, and optionally a set of negative prompts $Q^- \in \mathbb{R}^{N^- \times C}$, which in practise can be initialized from an open-set as above or with canonical phrases (e.g. 'object', 'thing' etc.) (Kerr et al., 2023). Similarity scores are converted to probabilities: $\mathcal{P} = \text{softmax}\left(\frac{1}{\gamma} \cdot \cos(\hat{Z}^{3D}, [\mathbf{q}^+, Q^-]^T)\right)$, where $\gamma$ a temperature hyper-parameter and $\mathcal{P} = [\boldsymbol{\rho}^+, \mathcal{P}^-]$ probabilities of positive matching $\boldsymbol{\rho}^+ \in \mathbb{R}^M$ and negative matching $\mathcal{P}^- = [\boldsymbol{\rho}_1^-, \ldots, \boldsymbol{\rho}_{N^-}^-] \in \mathbb{R}^{M \times N^-}$ respectively. The final 3D segmentation is given by $\hat{S}_i = \left(\boldsymbol{\rho}_i^+ > \max_j \mathcal{P}_{i,j}^-\right)$, or by thresholding $\boldsymbol{\rho}^+$ with a fixed threshold $s_{thr}$ (see ablations in Appendix A.3)

**Instance segmentation** Since our encoder has been distilled with the aid of instance-wise segmentation masks, the obtained features can be utilized out-of-the-box for 3D instance segmentation tasks. We demonstrate that with a simple clustering algorithm over $\hat{Z}^{3D}$ we can obtain 3D instance segmentation masks for cluttered scenes, where naive 3D coordinate clustering would fail, performing competitively with popular segmentation methods in unseen data in the single-view setting (see Sec. 4.3). We refer the reader to Appendix A.6.2 for implementation details and related visualizations.

| Point-Cloud | Features (PCA) | Ref@**cls** | Ref@**attribute** | Ref@**affordance** | Ref@**open** |
|---|---|---|---|---|---|
| | | book | potted plant | my child wants to play | Coke zero |
| | | hanger | round speaker | I need to write something | wall clock |
| | | headphones | yellow bottle | I have to check my email | toy bear |
| | | hat | pink flower | I'm hungry | LED monitor |

Figure 4: **Open-Vocabulary 3D Referring Segmentation in MV-TOD.** Examples of learned 3D features and grounding heatmaps from open-ended language queries (class names, attributes, user affordances, and open instance-specific concepts) in scenes from MV-TOD dataset. Points are colored based on their query similarity (higher towards red). We note that table points are excluded from similarity computation in our visualizations.

## 4 EXPERIMENTS

We design our experiments to explore the following questions: (i) **Sec. 4.1**: What are the contributions of our proposed object-centric priors for multi-view feature fusion? Does the dense number of views of our proposed dataset also contribute? (ii) **Sec. 4.2:** How does our method compare to state-of-the-art open-vocabulary approaches for semantic and referring segmentation tasks, both in multi- and in single-view settings? Is it robust to open-ended language? (iii) **Sec. 4.3:** What are the zero-shot generalization capabilities of our learned 3D representation in novel datasets that contain real-world scenes, as well as for the novel task of 3D instance segmentation? (v) **Sec. 4.4:** Can we leverage DROP-CLIP for language-guided 6-DoF robotic grasping?

### 4.1 MULTI-VIEW FEATURE FUSION ABLATION STUDIES

To evaluate the contributions of our proposed object-centric priors, we conduct ablation studies on the multi-view feature fusion pipeline, where we compare 3D referring segmentation results of obtained 3D features in held-out scenes of MV-TOD. We highlight that here we aim to establish a performance **upper-bound** that the feature fusion method can provide for distillation, and not the distilled features themselves. We ablate: (i) patch-wise vs. object-wise fusion, (ii) MaskCLIP (Dong et al., 2022) patch-level vs. CLIP (Radford et al., 2021) masked crop features,

| Fusion | f$^{2D}$ | $\Lambda_{v,i}$ | $G_{v,i}$ | mIoU | Pr@25 | Pr@50 | Pr@75 |
|---|---|---|---|---|---|---|---|
| | | | | | **Ref.Segm** (%) | | |
| point | patch | ✓ | | 37.3 | 55.4 | 33.7 | 16.7 |
| point | patch | | ✓ | 57.0 | 74.1 | 59.5 | 40.9 |
| point | patch | ✓ | ✓ | 57.4 | 77.0 | 60.9 | 39.9 |
| obj | obj | | | 65.6 | 67.0 | 65.4 | 64.1 |
| obj | obj | ✓ | | 67.3 | 68.7 | 67.1 | 65.8 |
| obj | obj | | ✓ | **83.1** | **83.9** | **83.1** | **82.4** |
| obj | obj | ✓ | ✓ | 80.9 | 83.1 | 80.2 | 79.7 |

Table 2: Multi-view feature fusion ablation study for 3D referring segmentation in MV-TOD.

(iii) inclusion of visibility ($\Lambda_{v,i}$) and semantic informativeness ($G_{v,i}$) metrics for view selection. We report 3D segmentation metrics *mIoU* and *Pr@X* (Wu et al., 2024). Results in Table 2.

**Effect of object-centric priors** We observe that all components contribute positively to the quality of the 3D features. Our proposed $G_{v,i}$ metric boosts *mIoU* across both point- and object-wise fusion (57.0% vs. 44.2% and 83.1% vs. 65.6% respectively). Further, we observe that the usage of spatial priors for object-wise fusion and object-level features leads to drastic improvements, both in segmentation crispness (25.7% *mIoU* delta), as well as in grounding precision (42.5% *Pr@75* delta).

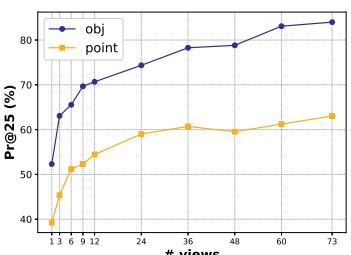

Figure 5: Referring segmentation precision vs. number of utilized views.

**Effect of the number of views** We ablate the 3D referring segmentation performance based on the number of input views in Fig. 5, where novel viewpoints are added incrementally. We observe that in both setups (point- and object-wise) fusing features from more views leads to

improvements, with a small plateauing behavior around 40 views. We believe this is an encouraging result for leveraging dense multi-view coverage in feature distillation pipelines, as we propose with MV-TOD. Please see Appendix A.3 for extended ablation studies that justify the design choices behind our fusion strategy, and Appendix A.5 for qualitative comparisons with vanilla approaches.

## 4.2 OPEN-VOCABULARY 3D SEGMENTATION RESULTS IN MV-TOD

In this section, we compare referring and semantic segmentation performance of our distilled features vs. previous open-vocabulary approaches, both in multi-view and in single-view settings.

For multi-view, we compare our trained model with OpenScene (Peng et al., 2022) and Open-Mask3D (Takmaz et al., 2023) methods, where the full point-cloud from all 73 views is given as input. We note that for these baselines we obtain the upper-bound 3D features as before, as we observed that our trained model already outperforms them, so we refrained from also distilling features from baselines. For single-view, we feed our network with partial point-cloud from projected RGB-D pair, and compare with 2D baselines MaskCLIP (Dong et al., 2022) and OpenSeg (Ghiasi et al., 2021) (see implementation details in Appendix A.4). Our model

| Method | #views | Ref.Segm. (%) | | | | Sem.Segm. (%) | |
|---|---|---|---|---|---|---|---|
| | | mIoU | Pr@25 | Pr@50 | Pr@75 | mIoU | mAcc$_{25}$ |
| OpenScene$^\dagger$ | 73 | 29.3 | 44.0 | 24.5 | 11.3 | 21.8 | 32.1 |
| OpenMask3D$^{*\dagger}$ | 73 | 65.4 | 73.1 | 64.0 | 57.4 | 59.5 | 66.5 |
| DROP-CLIP$^{*\dagger}$ | 73 | 82.7 | 86.1 | 82.4 | 79.2 | 75.4 | 80.0 |
| DROP-CLIP | 73 | 66.6 | 75.7 | 67.6 | 59.9 | 62.0 | 70.7 |
| OpenSeg$^{\rightarrow 3D}$ | 1 | 12.9 | 17.4 | 2.4 | 0.2 | 12.8 | 17.2 |
| MaskCLIP$^{\rightarrow 3D}$ | 1 | 25.6 | 40.4 | 18.7 | 7.0 | 21.0 | 32.1 |
| DROP-CLIP | 1 | 62.3 | 72.0 | 62.8 | 53.9 | 54.5 | 64.4 |

Table 3: *Referring* and *Semantic* segmentation results on MV-TOD test split. Methods with $^\dagger$ denote upper-bound 3D features, whereas DROP-CLIP denotes our distilled model. Methods with $^{\rightarrow 3D}$ produce 2D predictions that are projected to 3D to compute metrics. Methods with $*$ denote further usage of ground-truth segmentation masks.

slightly outperforms the OpenMask3D upper bound baseline in the multi-view setting ($+1.18\%$ in referring and $+2.57\%$ in semantic segmentation), while significantly outperforming 2D baselines in the single-view setting ($> 30\%$ in both tasks). Importantly, single-view results closely match the multi-view ones ($\sim -4.0\%$), suggesting that DROP-CLIP indeed learns view-independent features. See Appendix A.5 for more qualitative comparisons with baselines.

**Open-ended queries** We evaluate the robustness of our model in different types of input language queries, organized in 4 families (**class** name - e.g. *"cereal"*, **class + attribute** - e.g. *"brown cereal box"*, **open** - e.g. *"chocolate Kellogs"*, and **affordance** - e.g. *"I want something sweet'*). Comparative results are presented in Fig. 6 and qualitative in Fig. 4. We observe that single-view performance closely follows that of upper-bound across query types, with multi-word affordance queries being the highest family of failures, potentially due to the "bag-of-words" behavior of CLIP text embeddings (Shen et al., 2023).

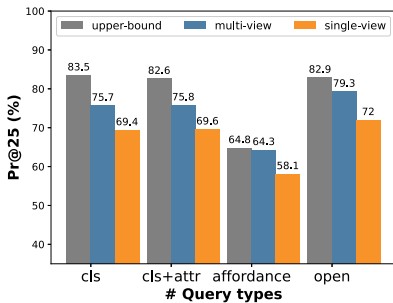

Figure 6: Referring segmentation precision vs. language query types.

## 4.3 GENERALIZATION TO NOVEL DOMAINS / TASKS

**Zero-shot transfer to real-world scenes** In this section, we evaluate the zero-shot generalization capability of DROP-CLIP in real-world scenes that contain objects and vocabulary outside the MV-TOD distribution. We test in the validation split of the OCID-VLG (Tziafas et al., 2023) dataset, which contains 1249 queries from 165 unique cluttered tabletop scenes. We compare with 2D CLIP-based baselines LSeg (Li et al., 2022a), OpenSeg (Ghiasi et al., 2021) and MaskCLIP (Dong et al., 2022) and popular 2D grounding method GroundedSAM (Ren et al., 2024) for the semantic segmentation task in the single-view setting as before.

Results are presented in Table 4. We find that even though fine-tuned in real data, baselines LSeg and OpenSeg under-perform compared to both MaskCLIP and our DROP-CLIP with a margin of $> 10\%$ mIoU, which we attribute to the distribution gap between the fine-tuning dataset ADE20K (Zhou et al., 2017) and OCID scenes. These baselines tend to ground multiple regions in the scene, while MaskCLIP and DROP-CLIP provides tighter segmentations (see Fig. 7). When considering the

| Method | OCID-VLG | | |
|---|---|---|---|
| | *mIoU* | *mAcc$_{50}$* | *mAcc$_{75}$* |
| GroundedSAM | 33.93 | 39.0 | 36.0 |
| LSeg$^{\rightarrow 3D}$ | 44.1 | 37.9 | 23.5 |
| OpenSeg$^{\rightarrow 3D}$ | 47.1 | 33.1 | 19.1 |
| MaskCLIP$^{\rightarrow 3D}$ | 57.1 | 59.4 | 31.0 |
| DROP-CLIP | 60.2 | 60.1 | 38.7 |

Table 4: Zero-shot semantic segmentation results (%) in the validation split of the OCID-VLG real-world dataset.

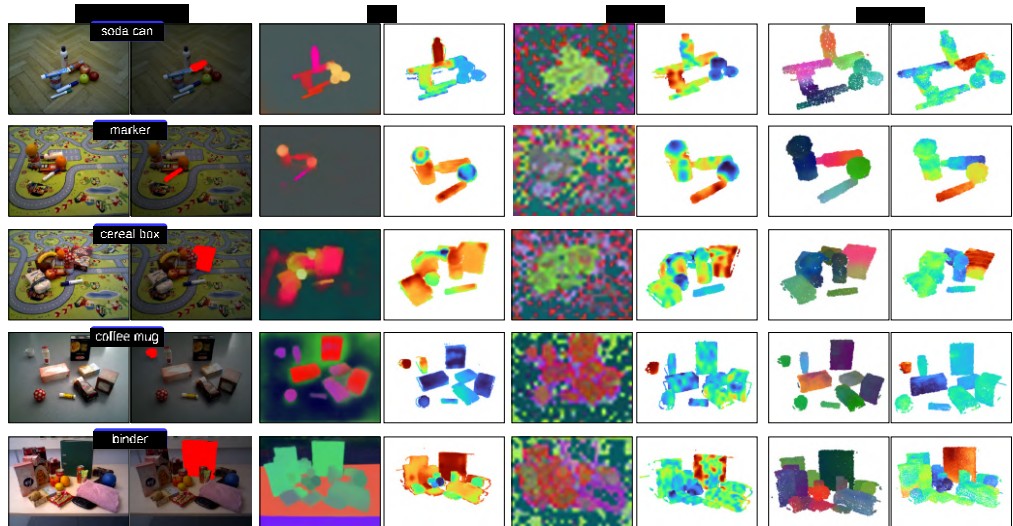

Figure 7: **Zero-Shot 3D Semantic Segmentation in Real Scenes:** Comparison of different referring segmentation models for five example cluttered indoor scenes from the OCID dataset. PCA features are displayed at pixel-level for 2D methods LSeg and MaskCLIP and in 3D for our point-cloud-based DROP-CLIP. Heatmaps from 2D models LSeg and MaskCLIP are projected to 3D for direct comparison with DROP-CLIP.

stricter $mAcc_{75}$ metric, our approach scores a delta of 7.7% compared to MaskCLIP, suggesting a significant gain in grounding accuracy, especially in cases where the object is heavily occluded. Failures cases were observed in grounding objects that significantly vary in geometry and semantics from the MV-TOD catalog. Please see Appendix A.6 for further zero-shot experiments, comparisons with modern NeRF/3DGS methods and more qualitative results.

**Zero-shot 3D instance segmentation** We evaluate the potential of DROP-CLIP for out-of-the-box 3D instance segmentation via clustering the predicted features (see details in Appendix A.6.2). We conduct experiments for both the multi-view setting in MV-TOD, where we compare with Mask3D (Schult et al., 2023) transferred from the ScanRefer (Chen et al., 2020) checkpoint provided by the authors, where we feed full point-clouds from 73 views, as well as in OCID-VLG, where we compare with SAM (Kirillov et al., 2023) ViT-L model with single-view images.

| Method | OCID-VLG | | MV-TOD | |
|---|---|---|---|---|
| | *mIoU* | $AP_{25}$ | *mIoU* | $AP_{25}$ |
| SAM | **60.1** | **95.3** | 70.1 | **95.2** |
| DROP-CLIP (S) | 50.9 | 68.0 | **80.8** | 91.9 |
| Mask3D | - | - | 14.4 | 18.7 |
| DROP-CLIP (F) | - | - | **88.3** | **93.3** |

Table 5: Zero-shot 3D instance segmentation results in OCID-VLG (real-world) and our MV-TOD dataset.

Results are summarized in Table 5. We observe that Mask3D struggles to generalize to tabletop domains, as it has been trained in room layout data with mostly furniture object categories. DROP-CLIP achieves an $AP_{25}$ of 93.3%, illustrating that the learned 3D features can provide near-perfect instance segmentation in-distribution, even without explicit fine-tuning. When moving out-of-distribution in the single-view setting, we observe that DROP-CLIP achieves *mIoU* that is competitive with foundation segmentation method SAM (50.9% vs. 60.1%). Failure cases include heavily cluttered regions of similar objects with same texture (e.g. food boxes), for which DROP-CLIP assigns very similar features that are identified as a single cluster.

## 4.4 APPLICATION: LANGUAGE-GUIDED ROBOTIC GRASPING

In this section, we wish to illustrate the applicability of DROP-CLIP in a language-guided robotic grasping scenario. We integrate our method with a 6-DoF grasp detection network (Chen et al., 2023), which proposes gripper poses for picking a target object segmented by DROP-CLIP. We randomly place 5-12 objects on a tabletop with different levels of clutter, and query the robot to pick a specific object, potentially amongst distractor objects of the same category. The user instruction is open-vocabulary

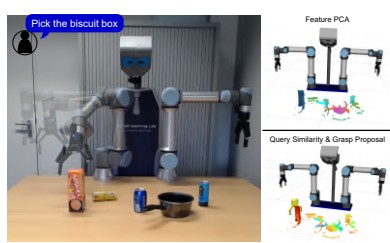

Figure 8: **Language-guided 6-DoF grasping**: Example robot trial *(left)*, 3D features, grounding and grasp proposal *(right)*.

and can involve open object descriptions, attributes, or user-affordances. We conducted 50 trials in Gazebo (Koenig & Howard, 2004) and 10 with a real robot, and observed grounding accuracy of 84% and 80% respectively, and a final success rate of 64% and 60%. Motion failures were mostly due to grasp proposals for which the motion planning led to collisions. Similar to OCID, grounding failures were due to unseen query concepts and / or instances. Example trials are shown in Fig. 8, more details in Appendix A.7 and a robot demonstration video is provided as supplementary material.

## 5 RELATED WORK

We briefly discuss related efforts in this section, while a detailed comparison is given in Appendix A.8.

**3D Scene Understanding** There's a long line of works in closed-set 3D scene understanding Choy et al. (2019); Han et al. (2020); Hu et al. (2021a;b); Li et al. (2022b); Robert et al. (2022), applied in 3D classification (Wu et al., 2014; Zhang et al., 2021), localization (Caesar et al., 2019; Chen et al., 2020) and segmentation (Behley et al., 2019; Ramakrishnan et al., 2021; Dai et al., 2017), using two-stage pipelines with instance proposals from point-clouds (Achlioptas et al., 2020a; Zhao et al., 2021) or RGB-D views (Huang et al., 2022; Liu et al., 2021), or single-stage methods (Luo et al., 2022) that leverage 3D-language cross attentions. (Rozenberszki et al., 2022a) use CLIP embeddings for pretraining a 3D segmentation model, but still cannot be applied open-vocabulary.

**Open-Vocabulary Grounding with CLIP** Following the impressive results of CLIP (Radford et al., 2021) for open-set image recognition, followup works transfer CLIP's powerful representations from image- to pixel-level (Gu et al., 2021; Zhong et al., 2021; Minderer et al., 2022; Zhou et al., 2022; Minderer et al., 2023; Wang et al., 2021; Lüddecke & Ecker, 2021; Ghiasi et al., 2021; Li et al., 2022a; Dong et al., 2022), extending to detection / segmentation, but limited to 2D. For 3D segmentation, the closest work is perhaps OpenMask3D (Takmaz et al., 2023) that extracts multi-view CLIP features from object proposals from Mask3D (Schult et al., 2023) to compute similarities with text queries.

**3D CLIP Feature Distillation** Recent works distill features from 2D foundation models with point-cloud encoders (Peng et al., 2022; Nguyen et al., 2023; Zhang et al., 2023) or neural fields (Kerr et al., 2023; Engelmann et al., 2024; Tschernezki et al., 2022; Kobayashi et al., 2022; Engelmann et al., 2024; Qin et al., 2024), with applications in robot manipulation (Rashid et al., 2023; Shen et al., 2023) and navigation (Shafiullah et al., 2022; Bolte et al., 2023). However, associated works extract 2D features from OpenSeg (Ghiasi et al., 2021), LSeg (Li et al., 2022a), MaskCLIP (Dong et al., 2022) or multi-scale crops from CLIP (Radford et al., 2021) and fuse point-wise with average pooling, while our approach leverages semantics-informed view selection and segmentation masks to do object-wise fusion with object-level features. Unlike all above field-based approaches, our method can be used real-time without the need for collecting multiple camera images at test-time.

## 6 CONCLUSION, LIMITATIONS & FUTURE WORK

We propose DROP-CLIP, a 2D→3D CLIP feature distillation framework that employs object-centric priors to select views based on semantic informativeness and ensure crisp 3D segmentations via leveraging segmentation masks. Our method is designed to work from single-view RGB-D, encouraging view-independent features via distilling from dense multi-view scene coverage. We also release MV-TOD, a large-scale synthetic dataset of multi-view tabletop scenes with dense semantic / mask / grasp annotations. We believe our work can benefit the community, both in terms of released resources as well as illustrating and overcoming theoretical limitations of existing 3D feature distillation works.

While our spatial object-centric priors lead to improved segmentation quality, they collapse local features in favor of a global object-level feature, and hence cannot be applied for segmenting object parts. In the future, we plan to add object part annotations in our dataset and fuse with both object- and part-level masks. Second, DROP-CLIP cannot reconstruct 3D features that have significantly different geometry and / or semantics from the object catalog used during distillation. In the future we aim to explore modern generative text-to-3D models to further scale up the object and concept variety of MV-TOD. Finally, regarding robotic application, currently DROP-CLIP only provides language grounding, and a two-stage pipeline is necessary for robot grasping, while MV-TOD already provides rich 6-DoF grasp annotations. A next step would be to also distill them, opting for a joint 3D representation for grounding semantics and grasp affordances.

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

# A    APPENDIX

## A.1    MV-TOD DETAILS

In this section we provide details for generating our MV-TOD scenes and their annotations (Sec. A.1.1) and present some statistics for the object and query catalog of MV-TOD (Sec. A.1.2).

### A.1.1    DATASET GENERATION

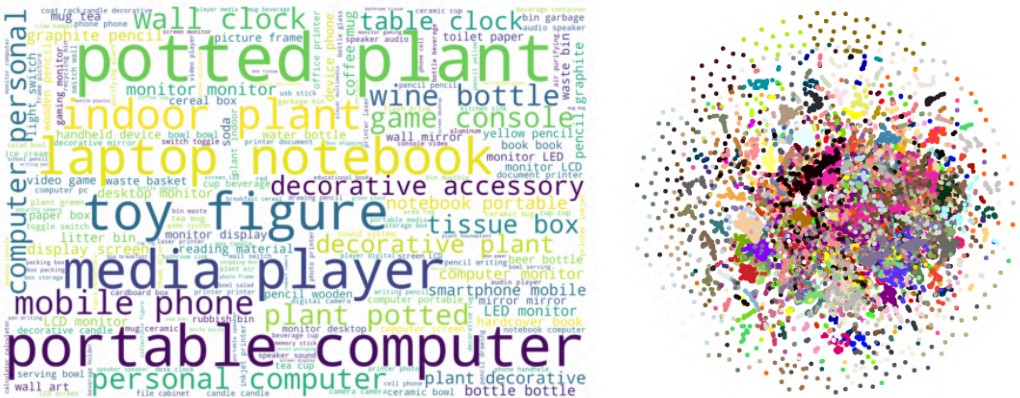

Figure 9: A wordcloud and T-SNE embedding projection visualization of textual concepts included in MV-TOD.

We generate the MV-TOD dataset in Blender (Community, 2018) engine with following steps:

**Random object spawn** For each scene, firstly, a support plane is spawned at the origin position. Then, random objects are selected to set up the multi-object tabletop scene. The number of objects ranges from 4 to 12, to make sure that our dataset covers both isolated and cluttered scenes. All selected objects are then spawned above the support plane with random position and random rotation. It is important to note that due to the limitation of Blender physical engine, an additional collision check is needed when an object is spawned into the scene to avoid initial collision. Since Blender does not provide users with the APIs to do the collision check, we check the collision by calculating the 3D IoU between object bounding boxes. After spawning object, the internal physical simulator is launched to simulation the falling of all the spawned object onto the plane. Once the objects are still, the engine will start rendering images.

**Multi-view rendering** In total 73 cameras are set in each scene for rending images from different views. One of them are spawned right on top of the origin position for rendering a top-down image, while the rest are uniformly distributed on the surface of the top hemisphere. An RGB image, a depth image (with the raw depth information in meters), and an instance segmentation mask are rendered at each view. All the annotations are saved in the COCO (Lin et al., 2014) JSON format for each scene.

**Data augmentation** In order to diversify the generated data, several augmentation methods are applied. Firstly, different textures and materials are randomly applied to the support plane, as well as the scene background, to simulate different types of table surfaces and background environments. Second, when the objects are spawned, their sizes and materials are randomly jittered. Thirdly, we also randomly slighlty modify the position of cameras towards the radial direction. Finally, the position and intensity of the light object in each scene are also randomly set.

**Semantic object annotation generation** To offer the functionality of querying target objects in our dataset by using high-level concepts and distinguish similar objects using fine-grained attributes, we also provide per-object semantic concepts generated with the aid of large vision-language models. For each object CAD model, we render 10 observation images from different views in Blender. Then, these images, together with an instruction prompt are fed to GPT-4V (GPT, 2023) to generate a response describing the current object in different perspectives, including category, color, material, state, utility, affordance, title (if applicable), and brand (if applicable). The text prompt we used to instruct GPT-4V is presented in Figure 10.

Please provide a set of text descriptions for the object shown in the input images. The input images are multiview observation of the object. The descriptions should describe the object color, material, state (e.g. if I give you the image of a bowl, you should tell if the bowl empty or full), as well as utility (e.g. if I give you the image of a hammer, you should say: "Something to do general carpentry, framing, nail pulling, cabinet making, assembling furniture, upholstering, finishing, riveting, bending or shaping metal, striking masonry drills and steel chisels, and so on"). Finally, provide a list of specific object descriptions that would be commonly used to refer to that object (e.g. If I give you an image of a coca-cola can, you could return: "[Coke, Coke-can, Coca-Cola, Cola, Cola-can, Cola-drink, ...]". If the object is a product, please try to identify and give its brand (e.g. "Coca-cola", "Fanta" etc.). Also provide a few `affordance` descriptions, which is what a user would say if they desired that object (e.g. If I give you an image of an apple, say: "I'm hungry", "I want to eat something healthy", etc.). Please reply with the following format:

**RESPONSE_FORMAT:**
---
**Category**: [give object category]
**Color**: [give object color]
**Material**: [give object material]
**State**: [give object state]
**Brand**: [optional - give product brand]
**Title**: [optional - give the title for object ONLY]
**Utility**: [give object utility]
**Affordance**: [give user instructions]
**More descriptions**: [give a list of specific object descriptions]
---
Please in your descriptions avoid words like "Appears to be [...]", "Appears [...]", "The object is [...]" etc., just give the noun/adjective descriptions.
Provide descriptions for the following **{label}** object

Figure 10: The text prompt we used for instructing GPT-4V. The {label} token will be replaced by the class name of the current object.

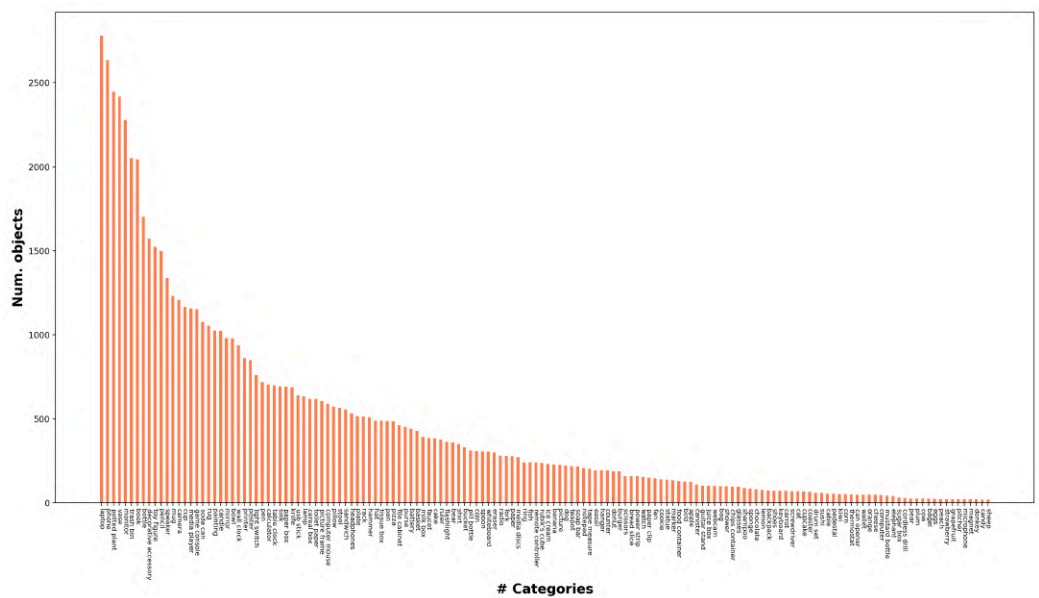

Figure 11: Number of objects in each category in MV-TOD.

**6-DoF grasp annotations** Since our model set originates from ShapeNet-Sem (Chang et al., 2015), we leverage the object-wise 6-DoF grasp proposals generated previously in the ACRONYM dataset (Eppner et al., 2020). These grasps were executed and evaluated in a simulation environment, leading to a total of 2000 grasp candidates per object. We filter the sucessfull grasps and connect them with each object instance in each of our scenes, by transforming the grasp annotation according to the recorded object's 6D pose from Blender. We further filter grasps by rendering a gripper mesh and removing all grasp poses that lead to collisions with the table or other objects.

### A.1.2 DATASET ANALYSIS

We visualize a wordcloud of the concept vocabulary of MV-TOD, together with tSNE projections of their CLIP text embeddings in Figure. 9. Certain object names (e.g. *"plant"*, *"computer"*, *"phone"*, *"vase"*) appear more frequently, as those are the objects that are most frequent in MV-TOD object catalog, hence they spawn a lot of expressions referring to them. Besides common class names, the wordcloud demonstrates that the most frequent concepts used to disambiguate objects are supplementary attributes (e.g. *decorative, potted, portable, etc*). Finally, colors and materials appear also frequently, as they are a common discriminating attribute between objects of the same category.

We further provide statistical analysis of MV-TOD in Table 6 and Figure 11. The number of referring expressions categorized by their types are listed in Table 6. We provide rich *open* expressions, which stems from open vocabulary concepts that can describe the referred objects in various aspects. As

it can be seen Figure 11, there exists a typical long-tail distribution in our dataset in terms of the number of objects per-category, where *laptop*, *phone*, and *plant* have the most variant instances.

| Type | Train | Test |
|------|-------|------|
| Class | 66.8k | 19.2k |
| Class+Attr | 76.5k | 21.8k |
| Affordance | 151.1k | 44.7k |
| Open | 356.8k | 102.1k |

Table 6: Number of referring expressions in MV-TOD organized by type

## A.2 DISTILLATION IMPLEMENTATION DETAILS

We use the `ViT-L/14@336px` variant of CLIP's vision encoder, which provides features of size $C = 768$ from $336 \times 448$ image inputs with patch size 14. We distill with a Minkowski-iNet14D (Choy et al., 2019) sparse 3D-UNet backbone, which consists of 8 sparse ResNet blocks with output sizes of $(32, 64, 128, 256, 384, 384, 384, 384)$ and a final $1 \times 1$ convolution head to 768 channels. To increase the 3D coordinates resolution, we upscale the input point-clouds to $\times 10$ and voxelize with original dimension of $d = 0.02$ (for feature fusion), and a voxel grid $d = 0.05$ for training with the Minkowski framework. To reduce the input dimensionality and speedup training and inference time, we remove the table points via filtering out the table's 3D mask. [1] We train using AdamW with initial learning rate $3 \cdot 10^{-4}$ and cosine annealing to $10^{-4}$ over 300 epochs, and a weight decay of $10^{-4}$. In each ResNet block, we include sparse batch normalization layers with momentum of 0.1. We train using two RTX 4090 GPUs, which takes about 4 days. Following (Peng et al., 2022; Takmaz et al., 2023) we use spatial

| Hyper-parameter | Value |
|-----------------|-------|
| voxel_size | 0.05 |
| feat_dim | 768 |
| color_trans_ratio | 0.01 |
| color_jitter_std | 0.02 |
| hue_max | 0.01 |
| saturation_max | 0.1 |
| elastic_distortion_granularity_min | 0.1 |
| elastic_distortion_granularity_max | 0.3 |
| elastic_distortion_magnitude_min | 0.4 |
| elastic_distortion_magnitude_max | 0.8 |
| n_blob_min | 1 |
| n_blob_max | 2 |
| blob_size_min | 50 |
| blob_size_max | 101 |
| random_euler_order | True |
| random_rot_chance | 0.6 |
| rotate_min_x | -0.1309 |
| rotate_max_x | 0.1309 |
| rotate_min_y | -0.1309 |
| rotate_max_y | 0.1309 |
| rotate_min_z | -0.1309 |
| rotate_max_z | 0.1309 |
| arch_3d | MinkUNet14D |
| batch_size | 8 |
| batch_size_val | 8 |
| base_lr | 0.0003 |
| weight_decay | 0.00001 |
| min_lr | 0.0001 |
| loss_type | cosine |
| use_aux_loss | False |
| use_cls_head | False |
| loss_weight_aux | 1.0 |
| loss_weight_cls | 0.1 |
| dropout_rate | 0.0 |
| epochs | 300 |
| power | 0.9 |
| momentum | 0.9 |
| max_norm | 5.0 |
| sync_bn | True |

Table 7: Training hyper-parameters

augmentations such as elastic distortion, horizontal flipping and small random translations and rotations. We also employ color-based augmentation such as chromatic auto-contrast, random color translation, jitter and hue saturation translation. To better emulate partial views with greater diversity, we train with full point-clouds but further add a per-object blob removal augmentation method that removes consistent blobs of points from each object instance. After training for 300 epochs, we fine-tune our obtained checkpoint on only partial point-clouds from randomly sampled views for each scene in our dataset. We experimented with several auxiliary losses to reinforce within-object feature similarity, such as supervised contrastive loss (Khosla et al., 2020) as well as KL triplet loss (Oki et al., 2020), but found that they do not significantly contribute to convergence compared to using only the main cosine distance loss. See Table 7 for a full overview of training and augmentation hyper-parameters.

---

[1]. During inference, we employ RANSAC to remove table points without access to segmentation masks.

## A.3 EXTENDED MULTI-VIEW FEATURE FUSION ABLATIONS

Our object-centric fusion pipeline considers several design choices besides the ones discussed in the main paper. In particular, we study: (i) Why masked crops as input to object-level 2D CLIP feature computation? How does it compare with other popular visual prompts to CLIP?, (ii) Why equations (3) and (4) in the semantic informativeness metric computation? How to sample negatives?, and (iii) What is the best strategy and hyper-parameters for doing inference?

**CLIP visual prompts** Previous works have extensively studied how to prompt CLIP to make it focus in a particular entity in the scene (Yang et al., 2023b; Shtedritski et al., 2023). We study the potential of visual prompting for obtaining object-level CLIP features in our object-centric feature fusion pipeline, via measuring their final referring segmentation *mIoU* in a subset of MV-TOD validation split. We compile the following

| crop | crop-mask | mask-blur | mask-gray | mask-out | #crops | crop-ratio | mIoU (%) |
|------|-----------|-----------|-----------|----------|--------|-----------|----------|
| ✗ |  |  |  |  | 1 | - | 81.9 |
| ✗ |  |  |  |  | 3 | 0.1 | 81.0 |
| ✗ |  |  |  |  | 3 | 0.15 | 80.6 |
| ✗ |  |  |  |  | 3 | 0.2 | 80.3 |
|  | ✗ |  |  |  | 1 | - | **84.0** |
|  | ✗ |  |  |  | 3 | 0.15 | 82.4 |
|  | ✗ |  |  |  | 3 | 0.2 | 82.0 |
| ✗ | ✗ |  |  |  | 1 | - | 81.7 |
|  |  | ✗ |  |  | - | - | 74.6 |
|  |  |  | ✗ |  | - | - | 57.4 |
|  |  |  |  | ✗ | - | - | 79.7 |
|  |  | ✗ | ✗ | ✗ | - | - | 70.2 |

Table 8: CLIP visual prompt ablation studies.

visual prompt options: (a) `crop`, where we crop a bounding box around each object (Kerr et al., 2023; Takmaz et al., 2023), (b) `crop-mask`, where we crop a bounding box but only leave the pixels of the object's 2D instance mask inside and uniformy paint the background (black, white or gray, based on the mask's mean color), (c) `mask-{blur,gray,out}`, where we use the entire image with the target object instance highlighted (Yang et al., 2023b) and the rest completely removed as before *(out)*, converted to grayscale *(gray)* or applied a median blur filter *(blur)*. For the crop options, we further ablate the number of multi-scale crops used and their relative expansion ratio. Results are summarized in Table 8. We observe the following: (a) image-level visual prompts used previously (Yang et al., 2023b) do not perform as well as cropped bounding boxes, (b) using multi-scale crops (Kerr et al., 2023; Takmaz et al., 2023) doesn't improve over using a single object crop, (c) masked crops outperform non-masked crops by a small margin of $2.1\%$. The difference is due to cases of heavy clutter, when the bounding box of the non-masked crop also includes neighboring objects, making the representation obtained by CLIP also give high similarities with the neighbor's prompt. This effect is more pronounced when using multiple crops with larger expansion ratios, as more and more neighboring objects are included in the crops.

**Semantic informativeness metric** We ablate the following components when computing semantic informativeness metric $G_{v,n}$: (i) the type of prompts used as $\mathbf{q}^+, Q^-$, i.e. `cls` for category-level prompts and `open` where we use all instance-level descriptions annotated with GPT-4V, (ii) the operator used to reduce the negative prompts to single feature dimension, i.e. `max` and `mean`, and (iii) how to sample negatives for $Q^-$, i.e. including only negative

| Prompts | Operator | Negatives | Ref.Segm. (%) | | Sem.Segm. (%) | |
|---------|----------|-----------|------|------|------|------|
| | | | mIoU | Pr@25 | mIoU | mAcc |
| cls | mean | scene | 82.2 | 83.1 | 73.1 | 75.1 |
| cls | max | scene | 82.8 | 84.0 | 74.9 | 76.7 |
| cls | mean | all | 80.9 | 81.0 | 71.6 | 73.9 |
| cls | max | all | 72.6 | 75.1 | 60.7 | 63.2 |
| open | mean | scene | 76.4 | 78.8 | 68.3 | 70.3 |
| open | max | scene | **83.9** | **85.5** | **75.6** | **77.2** |
| open | mean | all | 81.0 | 81.8 | 71.6 | 74.0 |
| open | max | all | 72.9 | 74.2 | 63.8 | 64.6 |

Table 9: Semantic informativeness metric ablation studies. Results in MV-TOD validation subset.

prompts for objects in the *scene*, or including *all* other dataset objects. Results are shown in Table 9. First, we observe that `max` operator generally outperforms *mean*, with the exception of when using *all* negatives. However, the best configuration was using `max` operator with *scene* negatives. Second, using `open` prompts provides marginal improvements over `cls` in all other settings. Finally, using *scene* negatives outperforms using *all* in most cases. This is because when using all negatives from the dataset, some semantic concepts will be highly similar with the positive prompt, making the metric too 'strict', as only few views will pass the condition $G_{v,n} \geq 0$.

**Inference strategies** As discussed in Sec. 3.4 there are two methods for performing referring segmentation inference: (a) selecting all points with higher probability for positive vs. maximum negative prompt $\rho^+ > \mathcal{P}^-$, or (b) thresholding $\rho^+$ with a hyper-parameter $s_{thr}$. Additionally, we compare the final referring segmentation performance based on the negative prompts used at test time: (a) prompts from object instances within the *scene*, (b) prompts from *all* dataset object instances (similar to se-

| Method | Negatives | Ref.Segm. (%) | | | |
|--------|-----------|------|-------|-------|-------|
| | | mIoU | Pr@25 | Pr@50 | Pr@75 |
| $\rho^+ > \mathcal{P}^-$ | scene | 73.7 | 77.4 | 73.0 | 69.8 |
| $\rho^+ > \mathcal{P}^-$ | canonical | 53.4 | 57.4 | 52.6 | 49.7 |
| $\rho^+ > \mathcal{P}^-$ | all | 30.8 | 31.0 | 31.0 | 30.8 |
| $s_{thr}$@0.95 | scene | **82.8** | **84.0** | **83.2** | **82.0** |
| $s_{thr}$@0.95 | canonical | 75.2 | 77.6 | 74.7 | 72.9 |
| $s_{thr}$@0.95 | all | 74.9 | 76.6 | 75.4 | 73.0 |
| $s_{thr}$@0.95 | - | 70.2 | 70.6 | 69.9 | 69.5 |
| $s_{thr}$@0.9 | scene | 82.1 | 83.6 | 82.8 | 79.8 |
| $s_{thr}$@0.8 | scene | 79.9 | 83.0 | 80.4 | 75.7 |

Table 10: Inference method ablation studies.

mantic segmentation task), (c) fixed *canonical* phrases {*"object", "thing", "texture", "stuff"*} (Kerr et al., 2023), and (d) no negative prompts (-), where we threshold the raw cosine similarities with the positive query. Results in Table 10. We observe that thresholding provides better results than the first method when the right threshold is chosen, a result which we found holds also for our distilled model. A high threshold of 0.95 was found optimal for upper bound experiments, while a threshold of 0.7 for our distilled model, although we further fine-tuned it for zero-shot and robot experiments (see Sec. A.6). Regarding negative prompts, as expected, providing in-scene negatives gives the best results, with a significant delta from canonical (7.6%), all (7.9%) and no negatives (12.6%). However, we observe that even without such prior, the performance is still competitive, even when entirely skipping negative prompts.

## A.4 BASELINE IMPLEMENTATIONS

**OpenSeg** (Ghiasi et al., 2021) extends CLIP's image-level visual representations to pixel-level, by first proposing instance segmentation masks and then aligning them to matched text captions. Given a text query, with OpenSeg we can obtain a 2D instance segmentation mask. For extending to 3D, we project the 2D mask pixels to 3D according to the mask region's depth values and camera intrinsics and transform to world frame.

**LSeg** (Li et al., 2022a) similarly trains an image encoder to be aligned with CLIP text embeddings at pixel-level with dense contrastive loss, therefore allowing open-vocabulary queries at test-time. Similar to OpenSeg, we project 2D predictions to 3D according to depth and camera intrinsics and transform to world frame to compute metrics.

**MaskCLIP** (Dong et al., 2022) provides a drop-in reparameterization trick in the attention pooling layer of CLIP's ViT encoder, enabling text-aligned patch features that can be directly used for grounding tasks. We use bicubic interpolation to upsample the patch-level features to pixel-level before computing cosine similarities with text queries. Similar to OpenSeg and LSeg, we project and transform the predicted 2D mask to calculate 3D metrics.

**OpenScene** (Peng et al., 2022) is the first method to introduce the 3D feature distillation methodology for room scan datasets. It utilizes OpenSeg (Ghiasi et al., 2021) to extract pixel-level 2D features and fuses them point-wise with vanilla average pooling, as formulated in Sec. 3.1. To provide fair comparisons with our approach, and as we found that MaskCLIP's features perform favourably vs. OpenSeg's, we use patch-wise MaskCLIP features, interpolated to original image size. We aggregate all 73 views, perform vanilla feature fusion in the full point-cloud, and measure the final fused 3D feature's performance as the OpenScene performance. We highlight that this setup represents the *upper-bound* performance OpenScene can provide, as we use the target 3D features and not distilled ones obtained through training, which we refrained from doing, as our results already outperform OpenScene's upper bound.

**OpenMask3D** (Takmaz et al., 2023) is a recent two-stage method for referring segmentation in point-cloud data. In the first stage, Mask3D (Schult et al., 2023) is used for 3D instance segmentation, providing a set of object proposals. In the second stage, multi-scale crops are extracted from rendered views around each proposed instance and passed to CLIP to obtain object-level features. For our implementation, similar to above, we wish to establish an upper-bound of performance OpenMask3D can obtain. To that end, we skip Mask3D in the first stage and provide ground-truth 3D segmentation masks. We represent each instance with a pooled CLIP feature from 3 multi-scale crops of 0.1 expansion ratio, obtained through all of our 73 views and weighted according to the visibility map $\Lambda_{v,n}$ (see Sec. 3.2), as in the original paper.

## A.5 QUALITATIVE RESULTS

We present qualitative results in several aspects to illustrate (1) How the object-centric priors help in multi-view feature fusion (Section A.5.1); (2) How do the distilled 3D features perform from single-view setting in MV-TOD semantic/referring segmentation tasks? (Section A.5.2).

### A.5.1 EFFECT OF OBJECT-CENTRIC PRIORS IN MULTI-VIEW FEATURE FUSION

We present more visualizations to demonstrate the difference between our method and previous multi-view feature fusion approaches, highlighting the effectiveness of injecting object-centric priors in

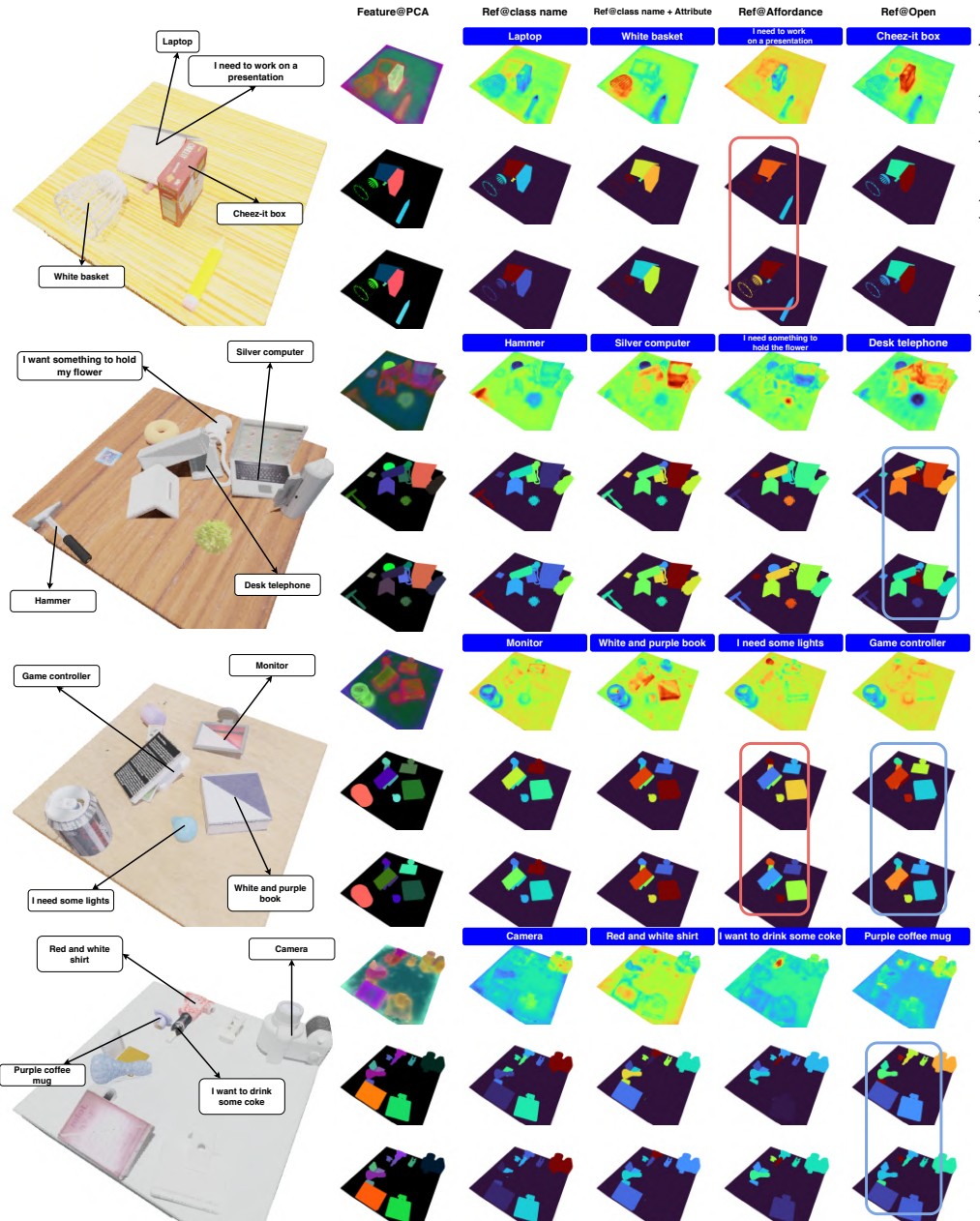

Figure 12: PCA feature and referring grounding visualization of baseline methods and DROP-CLIP. For each scene, we present results for OpenScene, OpenMask3D, and our DROP-CLIP (from top to bottom). The blue rectangle denotes cases where OpenMask3D suffers from distractor objects, while DROP-CLIP doesn't. The red rectangle denotes cases where OpenMask3D totally fails to ground the target, while DROP-CLIP succeeds.

fusing process. The results in Fig. 12 and Fig. 13 show the upper bound features of OpenScene(Peng et al., 2022), OpenMask3D (Takmaz et al., 2023), and our DROP-CLIP. It can be seen that by introducing the segmentation mask spatial priors, both OpenMask3D and DROP-CLIP can obtain more crispy features in the latent space and also achieve better language grounding results. To demonstrate the benefit of introducing the semantic informativeness metric in feature fusion, we add extra annotation in Fig. 12 and Fig. 13. The blue rectangle denotes the cases where OpenMask3D suffers from the distractors (i.e. multiple objects have high similarity score with the given query), while our DROP-CLIP is not. The red rectangle denotes the cases where OpenMask3D totally failed to ground the correct object, while our DROP-CLIP succeed. In conclusion, introducing semantic informativeness results in more robust object-level embeddings that in turn lead to higher grounding accuracy.

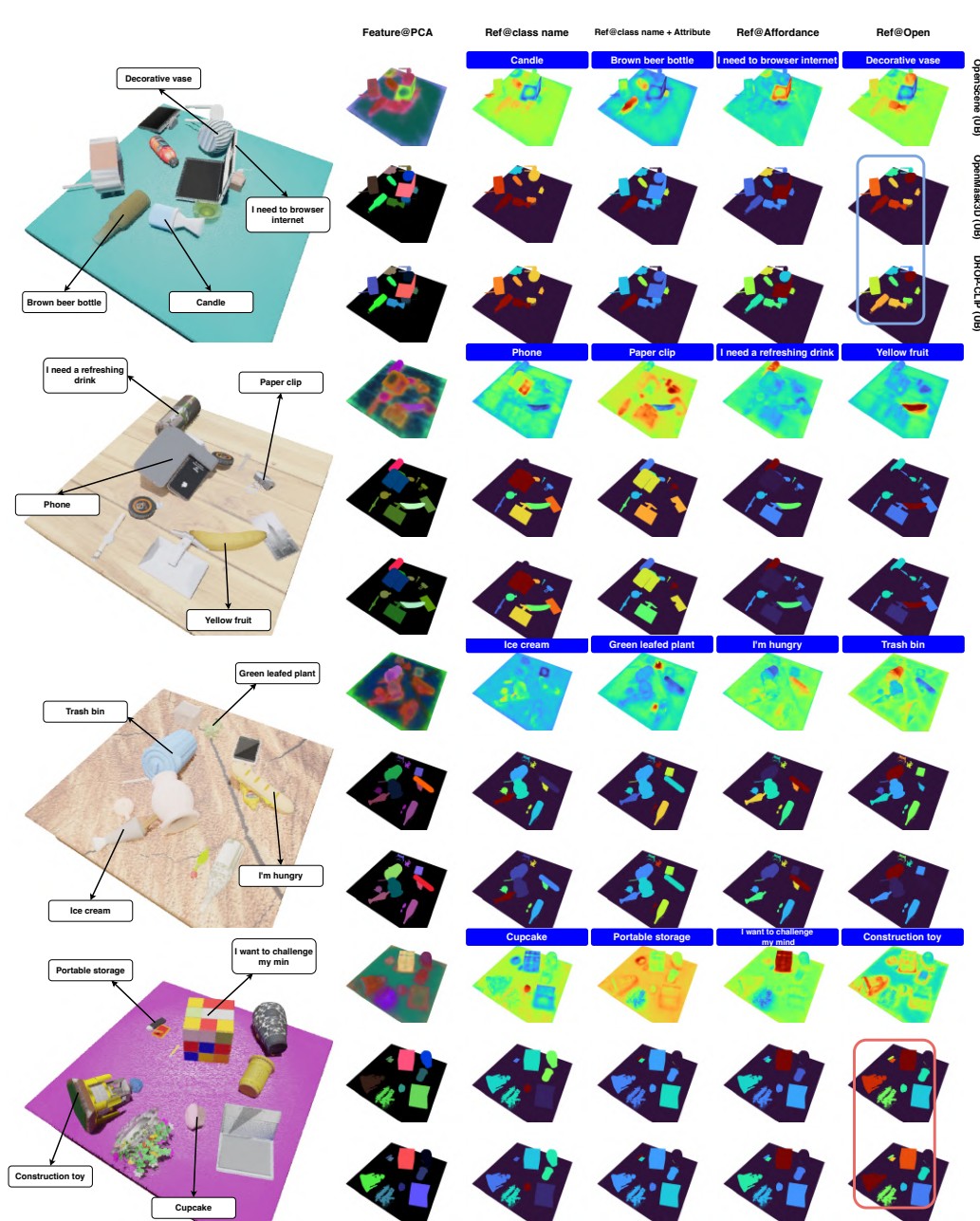

Figure 13: PCA feature and referring grounding visualization of baseline methods and DROP-CLIP. For each scene, we present results for OpenScene, OpenMask3D, and our DROP-CLIP (from top to bottom). The blue rectangle denotes cases where OpenMask3D suffers from distractor objects, while DROP-CLIP doesn't. The red rectangle denotes cases where OpenMask3D totally fails to ground the target, while DROP-CLIP succeeds.

### A.5.2 REFERRING / SEMANTIC SEGMENTATION QUALITATIVE RESULTS

**Referring segmentation** Since DROP-CLIP is not trained on closed-set vocabulary dataset but rather to reconstruct the fused multi-view CLIP features, the distilled features naturally live in CLIP text space. As a result, we can conduct referring expression segmentation in 3D with open vocabularies. We demonstrate this ability in Fig. 14 by showing the grounding results of the trained DROP-CLIP queried with different language expression, including *class name*, *class name + attribute*, *affordance*, and *open* instance-specific queries.

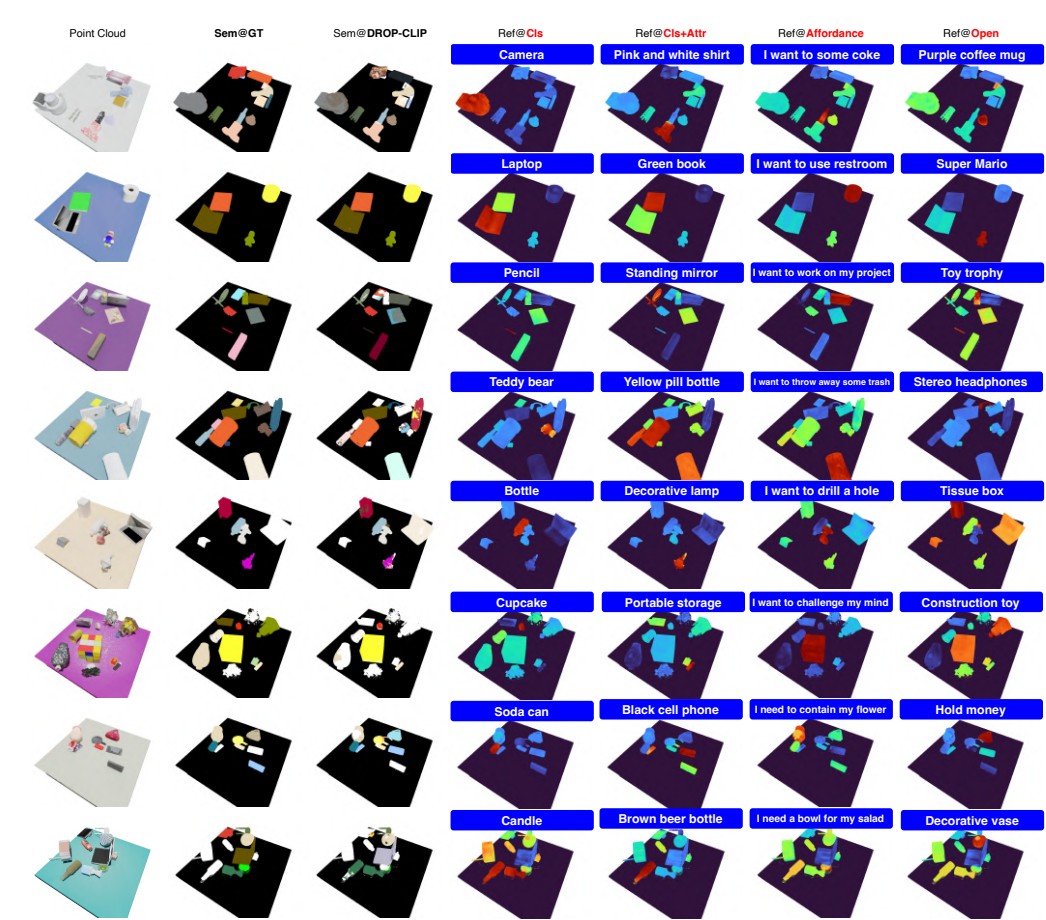

Figure 14: Semantic/Referring segmentation with our DROP-CLIP. In the **Sem@** columns, the same colors denote the same object category. The white parts mean that this part of the object is not activated by the corresponding class name query.

**Semantic segmentation** We present semantic segmentation results of our DROP-CLIP in Fig. 14. The white parts in Fig. 14 mean that this part of the object is not activated by the corresponding class name query.

## A.6 ZERO-SHOT TRANSFER EXPERIMENTS DETAILS

To study the transferability of our learned 3D features in novel tabletop domains, in Sec. 4.3 we conducted single-view semantic segmentation experiments in the OCID-VLG dataset. In this setup, similar to our single-view MV-TOD experiments, we project the input RGB-D image to obtain a partial point-cloud and feed it to DROP-CLIP to reconstruct 3D CLIP features. To represent the point-clouds in the same scale as our MV-TOD training scenes, we sweep over multiple scaling factors and report the ones with the best recorded performance. For 2D baselines, the *mIoU* and *mAcc@X* metrics were computed based on the ground-truth 2D instance segmentation masks of each scene, after projected to 3D with the depth image and camera intrinsics and transformed to world frame, fixed at the center of the tabletop of each dataset.

### A.6.1 ZERO-SHOT REFERRING SEGMENTATION EXPERIMENTS

Since methods LSeg and OpenSeg were fine-tuned for semantic segmentation, they are not suitable for grounding arbitrary referring expressions, but only category names as queries, which is why we conducted semantic segmentation experiments in our main paper. To further study zero-shot referring

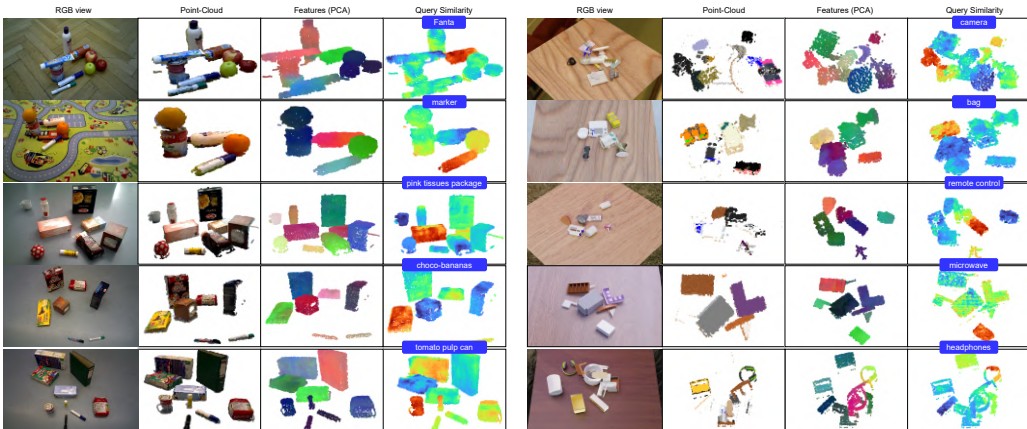

Figure 15: Visualization of referring segmentation examples in OCID-VLG *((left)* and REGRAD *(right)* datasets.

segmentation generalization, we conducted additional experiments in both OCID-VLG Tziafas et al. (2023) and REGRAD Zhang et al. (2022) datasets. We compare with the MaskCLIP baseline, projected to 3D similar to above. For both the MaskCLIP baseline and DROP-CLIP, we use thresholding inference strategy, sweep over thresholds $\{0.4, \ldots, 0.9\}$ and record the best configuration for both methods. We analyze the utilised datasets below:

**OCID-VLG** (Tziafas et al., 2023) connects 4-DoF grasp annotations from OCID-Grasp (Ainetter & Fraundorfer, 2021) dataset with single-view RGB scene images and language data generated automatically with templated referring expressions. We evaluate in one referring expression per scene for a total of 490 scenes, 165 from the validation and 325 from the test set of the *unique* split provided by the authors. We use the dataset's referring expressions from *name* type as queries, after parsing out the verb (i.e. *"pick the"*), which contains open descriptions for 58 unique object instances, incl. concepts such as brand, flavor etc. (e.g. *"Kleenex tissues"*, *"Choco Krispies corn flakes"*, *"Colgate"*). For removing the table points, we use the provided ground-truth 2D segmentation mask to project only instance points to 3D for DROP-CLIP. We sweep over scaling factors $\{8, \ldots, 16\}$ in the validation set and report the best obtained results.

**REGRAD** (Zhang et al., 2022) focuses on 6-DoF grasp annotations and manipulation relations for cluttered tabletop scenes. Scenes are rendered from a pool of $50k$ unique ShapeNet (Chang et al., 2015) 3D models from 55 categories with 9 RGB-D views from a fixed height. We test in $1000$ random scenes from *seen-val* split, using ShapeNet category names as queries. We note that as REGRAD doesn't focus on semantics but grasping, most of its objects are not typical household objects, but furniture objects (e.g. tables, benches, closets etc.) scaled down and placed in the tabletop. We filter out queries with such object instances and experiment with the remaining 16 categories that represent household objects (e.g. *"bottle", "mug", "camera"* etc.). We sweep over scaling factors $\{6, \ldots, 20\}$. We use the filtered full point-clouds provided by the authors to identify the table points and remove them from each view.

More qualitative results for both datasets are illustrated in Fig. 15, while comparative results with MaskCLIP are given in Table 11. We observe that our method provides a significant performance boost across both domains ($5.8\%$ *mIoU* delta in OCID-VLG and $25.9\%$ in REGRAD), especially in REGRAD scenes, where objects mostly miss fine texture and have plain colors, thus leading to poor

| Method | OCID-VLG | | REGRAD | |
|---|---|---|---|---|
| | mIoU | Pr@25 | mIoU | Pr@25 |
| MaskCLIP$^{\rightarrow 3D}$ | 40.4 | 45.2 | 33.2 | 39.0 |
| DROP-CLIP | **46.2** | **48.9** | **59.1** | **63.0** |

Table 11: Zero-shot referring segmentation results in OCID-VLG and REGRAD datasets

MaskCLIP predictions compared to DROP-CLIP, which considers the 3D geometry of the scene. Failures were observed in cases of very unique referring queries in OCID-VLG (e.g. *"Keh package"*) and in cases of very heavily occluded object instances in both datasets.

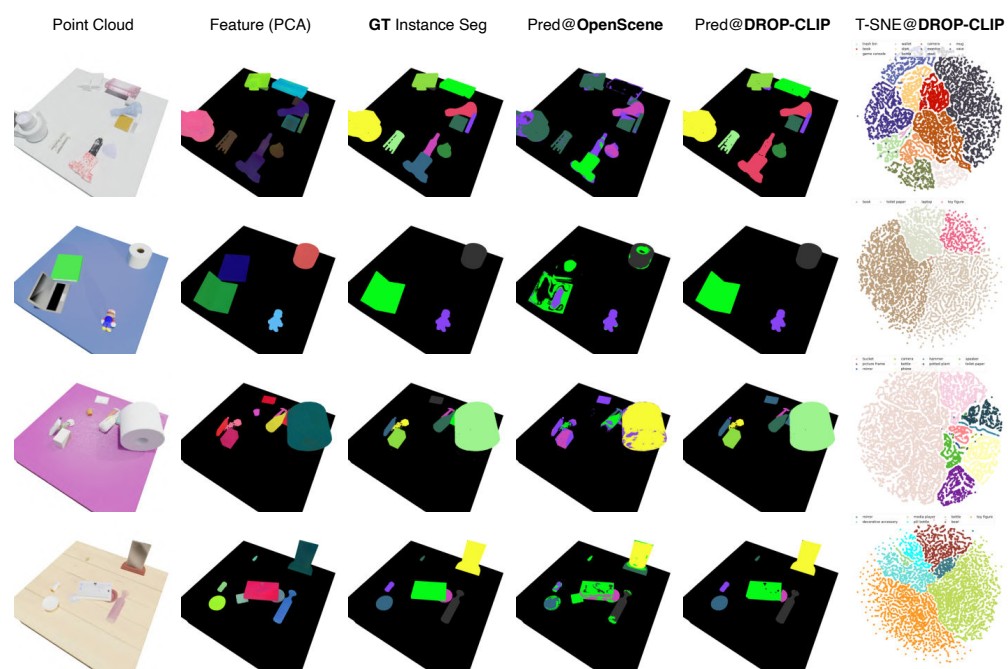

Figure 16: Zero-shot instance segmentation with our DROP-CLIP. In the *GT* and *Pred* columns, the same colors denote the same instance.

### A.6.2 ZERO-SHOT 3D INSTANCE SEGMENTATION EXPERIMENTS

Integrating spatial object priors via segmentation masks when fusing multi-view features grants separability in the embedding space. To illustrate that, we conducted zero-shot instance segmentation with our DROP-CLIP by directly applying DBSCAN clustering in the output feature space. In our experiments, we use the vanilla implementation of DBSCAN from `scikit-learn` package and set $\epsilon = 0.01$, `min_samples` $= 2$ for DROP-CLIP and $\epsilon = 0.01$, `min_samples` $= 276$ for OpenScene respectively. We observed that the points that belong to the same object instance are close to each other in the feature space, while significantly differ from the points that belong to other instances. We visualize several examples in Fig. 16, where we also conduct a t-SNE visualization to demonstrate the instance-level separability in the DROP-CLIP feature space.

### A.6.3 COMPARISONS WITH SfM METHODS

In this section we compare DROP with modern 2D→3D feature distillation methods based on *Structure-from-Motion (SfM)*, obtained via training NeRFs (Kerr et al., 2023; Engelmann et al., 2024; Shen et al., 2023; Kobayashi et al., 2022) or 3D Gaussian Splatting (3DGS) (Qin et al., 2024; Guo et al., 2024; Qiu et al., 2024; Zhou et al., 2023). We highlight however that this is not really an "apples to apples" comparison, since SfM approaches differ from our method in philosophy and scope of application. In particular, SfM approaches perform **online** distillation

| Method | Modality | Num. Views | Train Time | Segm. Model | Results | |
|---|---|---|---|---|---|---|
| | | | | | Loc. | Sem.Segm. |
| LERF | SfM | 171 | 112.5 min. | - | 84.8 | 45.0 |
| LangSplat | SfM | 171 | 37.5 min. | SAM | 88.1 | 65.1 |
| SemanticGaussians | SfM | 171 | >2 hrs. | SAM | 89.8 | - |
| LSeg | RGB | 1 | 0 | - | 33.9 | 21.7 |
| DROP-CLIP | RGB-D | 1 | 0 | - | 66.1 | 39.1 |

Table 12: Localization accuracy (%) and 3D semantic segmentation mIoU (%) on the *'teatime'* scene of LERF dataset. We report number of views, training time and whether / which external models are needed to obtain the representation. Training times are converted to v100 hours from reported numbers in corresponding papers.

in specific scenes, and thus require multiple camera images to distill, as well as significant time to do training / inference. The obtained scene representation cannot be applied in new scenes, for which a new multi-view images dataset has to be constructed and a new NeRF / 3DGS be trained from scratch. In contrast, our approach relies on depth sensors to acquire 3D and does not need SfM reconstruction. The feature distillation is performed **offline once**, in the MV-TOD dataset, and thus

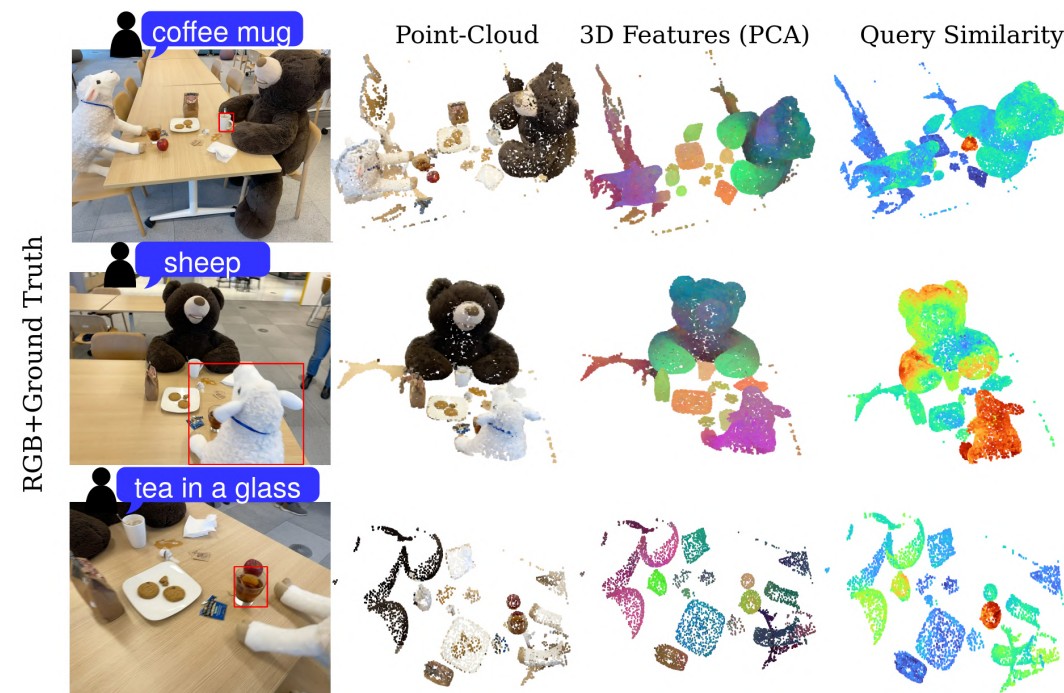

Figure 17: Visualizations of partial point-clouds, 3D DROP-CLIP features (PCA) and similarity heatmaps for three different queries in the *'teatime'* scene of LERF dataset.

can be applied zero-shot in novel scenes. Further, it does not require multiple camera images (works from single-view), does not require training and supports real-time inference. Nevertheless, we want to quantify the relative performance of DROP-CLIP with SfM methods that have been distilled for specific scenes.

We replicate the setup of the localization task from LERF (Kerr et al., 2023) and the semantic segmentation task from LangSplat (Qin et al., 2024) for the *'teatime'* scene of the LERF dataset. Results are presented in Table 12, where numbers for representative baselines LSeg (Li et al., 2022a), LERF (Kerr et al., 2023), LangSplat (Qin et al., 2024) and Semantic Gaussians (Guo et al., 2024) are taken from corresponding papers. To signify the aforementioned differences in scope, in our table we also report number of views and training time required to obtain the representation (converted in v100 hours from time reported in corresponding papers) and whether / which external segmentation models (e.g. SAM (Kirillov et al., 2023)) is needed during test-time to deal with the 'patchyness' issue. The above demonstrate the practical benefits of our approach compared to SfM methods, as mentioned before, working from single-view, real-time performance, zero-shot application and no need for external segmentors. Regarding test results, we find that DROP scores lower to SfM baselines in both task variants, but significantly outperforms LSeg, which is the only other zero-shot baseline. The performance margin between DROP and object-centric 3DGS methods LangSplat and SemanticGaussians is significant, albeit the fact that these methods require SAM at test-time to inject the segmentation priors, whereas DROP doesn't. This gap is justified when considering that our approach is zero-shot and didn't have access to the 171 training scenes like the SfM baselines, as well as that the dataset queries are often referring to object parts (e.g. *hooves*, *bear nose* etc.), which DROP has not been designed for. Qualitative visualizations of DROP in the LERF scene are given in Fig. 17.

### A.7 ROBOT EXPERIMENTS

**Setup** Our robot setup consists of two UR5e arms with Robotiq 2F-140 grippers and an ASUS Xtion depth camera mounted from an elevated view between the arms. We conducted 50 trials in the Gazebo simulator (Koenig & Howard, 2004) and 10 with a real robot. For simulation, we used 29 unique object instances from 9 categories (i.e. soda cans, fruit, bowls, juice boxes, milk boxes,

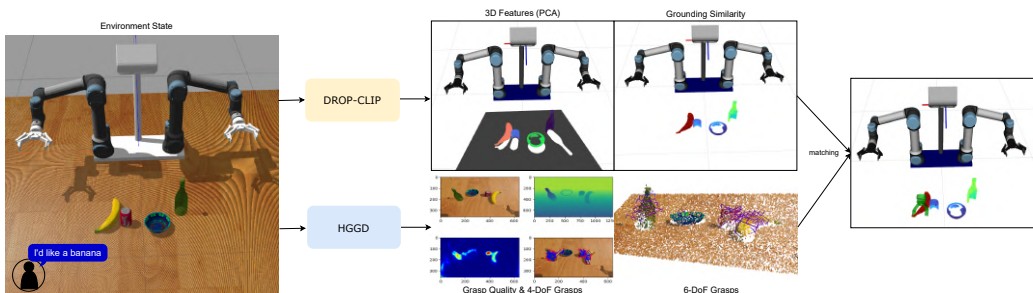

Figure 18: Illustration of robot system for language-guided 6-DoF grasping, using our DROP-CLIP for grounding *(top)*, and HGGD network (Chen et al., 2023) for grasp detection *(bottom)*.

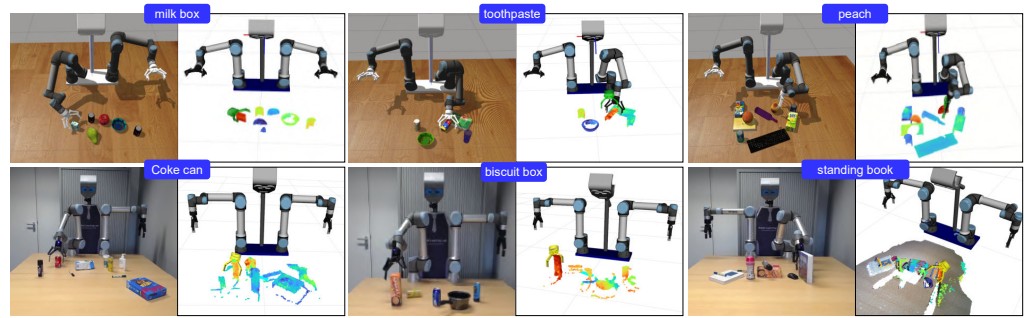

Figure 19: Visualization of robot experiments in Gazebo *(top)* and with a real robot *(bottom)*.

bottles, cans, books and edible products). For real robot experiments, we mostly used packaged products and edibles. In each trial, we place 5-12 objects in a designated workspace area. Objects are either scattered across the workspace, packed together in the center or partially placed in the same area in order to emulate different levels of clutter. We provide a query indicating a target object using either category name, color/material/state attribute, user affordance (e.g. *"I'm thirsty"*), or open instance-level description, typically referring to the object's brand (e.g. *"Pepsi", "Fanta"* etc.) or flavor (e.g. *"strawberry juice", "mango juice"* etc.) We note that distractor object instances of the same category as the target object are included in trials where query is not the category name.

**Implementation** We develop our language-guided grasping behavior in ROS, using DROP-CLIP for grounding the user's query and RGB-D grasp detection network, `HGGD` (Chen et al., 2023), for generating 6-DoF grasp proposals. Our pipeline is shown in Fig. 18. We process the raw sensor

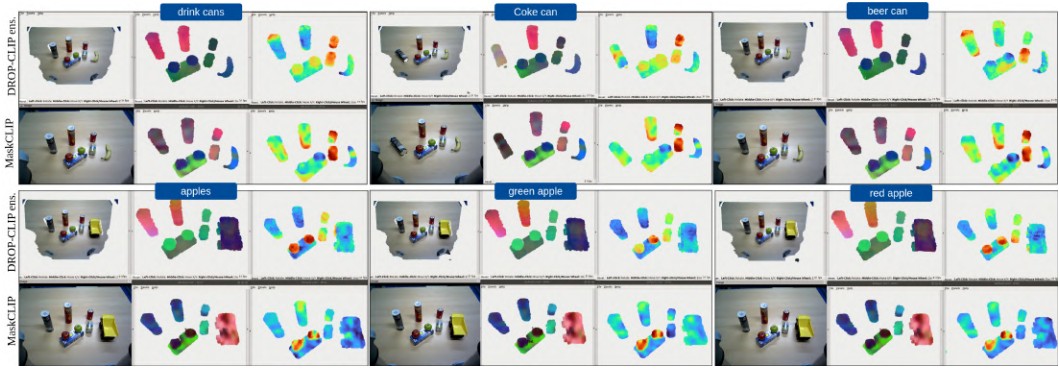

Figure 20: Visualization of grounding queries in real robot trials, for baseline method MaskCLIP$^{\rightarrow 3D}$ *(bottom)* and our DROP-CLIP *(top)*, where we ensemble the predictions of our method with the 2D baseline. DROP-CLIP produces more robust features *(middle column)* which lead to crispier segmentation *(right column.)*

point-cloud with RANSAC from `open3d` library with distance threshold 0.1, `ransac_n`=3 and 1000 iterations to segment out the table points, and then upscale to ×10. We use in-scene category names as negative prompts and do inference with a threshold of 0.7. To match the grounded object points with grasp proposals, we transform predicted grasps to world frame and move their center at the gripper's tip. We then calculate euclidean distances between the gripper's tip and the thresholded prediction's center. In real robot experiments, we run statistical outlier removal from `open3d` with `neighbor_size=25` and `std_ratio=2.0` to remove noisy points from the prediction's center. The top-3 closest grasps are given as goal for an inverse kinematics motion planner. We manually mark grasp attempts as success/failure in real robot and leverage the simulator state to do it automatically in Gazebo. Visualizations of simulated / real robot trials are illustrated in Fig. 19, experiments with grounding different objects with fine-grained attributes in Fig. 20, while related videos are included as supplemetary material.

### A.8    DETAILED RELATED WORK

In this section we provide a more comprehensive overview of comparisons with related work.

**Semantic priors for CLIP in 3D** A line of works aim to learn 3D representations that are co-embedded in text space by leveraging textual data, typically with contrastive losses (Ding et al., 2022; Yang et al., 2023a; Ding et al., 2023). CG3D (Hegde et al., 2023) aims to learn a multi-modal embedding space by applying contrastive loss on 3D features from point-clouds and corresponding multi-view image and textual data, while using prompt tuning to mitigate the 3D-image domain gap. Most above methods lead to a degradation in CLIP's open-vocabulary capabilities due to the fine-tuning stages. In contrast, our work leverages textual data not for training but for guiding multi-view visual feature fusion, hence leaving the learned embedding space intact from CLIP pretraining.

**Spatial priors for CLIP in 3D** Several works propose to leverage spatial object-level information to guide CLIP feature computation in 3D scene understanding context. OpenMask3D (Takmaz et al., 2023) leverages a pretrained instance segmentation method to provide object proposals, and then extracts an object-level feature by fusing CLIP features from multi-scale crops. Similarly, OpenIns3D (Huang et al., 2023) generates object proposals and employs a Mask-Snap-Lookup module to utilize synthetic-scene images across multiple scales. In similar vein, works such as Open3DIS (Nguyen et al., 2023), OVIR-3D (Lu et al., 2023), SAM3D (nuo Yang et al., 2023), MaskClustering (Yan et al., 2024) and SAI3D (Yin et al., 2023) leverage pretrained 2D models to generate 2D instance-wise masks, which are then back-projected onto the associated 3D point cloud. All above approaches are two-stage approaches that rely on the instance segmentation performance of the pretrained model in the first stage, thus suffering from cascading effects when segmentations are not accurate or well aligned across views. In contrast, our method leverages spatial priors *during* the multi-view feature fusion process, and then distills the final features with a point-cloud encoder, and therefore is a single-stage method that does not require object proposals at test time.

**Offline 3D CLIP Feature Distillation** OpenScene (Peng et al., 2022) distills OpenSeg (Ghiasi et al., 2021) multi-view features with a point-cloud encoder, while follow-up work Open3DSG (Koch et al., 2024) extends to scene graph generation by further distilling object-pair representations from other vision-language foundation models (Dai et al., 2023) as graph edges. CLIP-FO3D (Zhang et al., 2023) replaces OpenSeg pixel-wise features with multi-scale crops from CLIP to further enhance generalization. All above works use dense 2D features and fuse point-wise, thus suffering from 'patchyness' issue. Further, these works distill features using 3D room scan data (Dai et al., 2017; Chen et al., 2020), which lack diverse object catalogs and do not have to deal with the effects of clutter in the multi-view fusion process, as we do with the introduction of MV-TOD.

**Online 3D CLIP Feature Distillation** LERF (Kerr et al., 2023) replaces point-cloud encoders with neural fields, and distils multi-scale crop CLIP features into a continuous feature field that can provide features in any region of the input space. The authors deal with the 'patchyness' issue using DINO regularization. Similar works OpenNeRF (Engelmann et al., 2024) and F3RM (Shen et al., 2023) use MaskCLIP to extract features and avoid DINO-regularization. All above works make the assumption that all views are equally informative and rely on dense number of views at test-time to resolve the noise in the distilled features. A more recent line of works replace NeRFs with 3D Gaussian Splatting (3DGS) (Kerbl et al., 2023) to improve inference time and memory consumption and perform similar feature distillation from LSeg, OpenSeg or CLIP multi-scale crops. Similar to our

work, some 3DGS approaches (Qin et al., 2024; Guo et al., 2024; Qiu et al., 2024; Zhou et al., 2023) also exploit spatial priors (i.e. segmentation masks) to distill object-level CLIP features, but do not perform view selection based on semantics. Further, 3DGS approaches lie in the same general family of works as fields, i.e., online distillation in specific scenes, requiring multiple camera images and computational resources to work at test-time. In contrast, our method is distilled offline in MV-TOD to reconstruct semantically-informed, view-independent 3D features from single-view RGB-D inputs, can be applied zero-shot in novel scenes without training, and enables real-time inference.

