# OpenReview forum: "View-Independent 3D Feature Distillation with Object-Centric Priors"
_ICLR.cc/2025/Conference — Submitted to ICLR 2025_

### Official Review · Reviewer_n2dh · 2024-11-02

**Soundness:** 2
**Presentation:** 2
**Contribution:** 2
**Rating:** 5
**Confidence:** 5

**Summary:**

The paper addresses the limitations of 3D feature distillation in previous methods, which assumed all camera views are equally informative. To tackle this, it introduces MV-TOD, a synthetic dataset with detailed 3D masks and 6DoF grasp pose annotations for tabletop scenes. The paper then proposes DROP-CLIP, a method that employs object-centric priors for weighted multi-view feature re-projection. Additionally, to simplify the application for robotics, it introduces view-independent feature distillation, enabling the extraction of 3D open-vocabulary features from a single view. Experimental results demonstrate the method's effectiveness, including its utility in language-guided (open-vocabulary) robotic grasping.

**Strengths:**

1. The method demonstrates strong results in CLIP/DINO feature distillation and performs well on object-level segmentation.
2. The experiments are comprehensive, with sufficient ablation studies provided.
3. The paper is well-written and clearly communicates its ideas.

**Weaknesses:**

1. A broader set of baselines should be included in experiments, such as LERF [1], D3Field [2], and Semantic Gaussians [3].
2. The MV-TOD dataset, while useful, is synthetic and introduces a sim-to-real gap when using modified CLIP/DINO for dense feature extraction, which limits its impact.
3. For open-vocabulary grasping, comparing the proposed 3D representation to baselines like F3RM [4], LERF-TOGO [5], and GaussianGrasper [6] would help validate its advantages.
4. The novelty of the method may fall short for an ICLR submission, as it relies on projecting 2D CLIP or DINO features into 3D points via simple averaging, which is not a new approach.
5. The results from real-world scenes (Fig. 7) are not particularly impressive; comparisons with recent methods, such as Semantic Gaussians [3] and GaussianGrasper [6], would strengthen the evaluation.

[1] Kerr, Justin, et al. "Lerf: Language embedded radiance fields." Proceedings of the IEEE/CVF International Conference on Computer Vision. 2023.

[2] Wang, Yixuan, et al. "D3Fields: Dynamic 3D Descriptor Fields for Zero-Shot Generalizable Robotic Manipulation." arXiv preprint arXiv:2309.16118 (2023).

[3] Guo, Jun, et al. "Semantic Gaussians: Open-Vocabulary Scene Understanding with 3D Gaussian Splatting." arXiv preprint arXiv:2403.15624 (2024).

[4] Shen, William, et al. "Distilled feature fields enable few-shot language-guided manipulation." arXiv preprint arXiv:2308.07931 (2023).

[5] Rashid, Adam, et al. "Language embedded radiance fields for zero-shot task-oriented grasping." 7th Annual Conference on Robot Learning. 2023.

[6] Zheng, Yuhang, et al. "GaussianGrasper: 3D Language Gaussian Splatting for Open-vocabulary Robotic Grasping." arXiv preprint arXiv:2403.09637 (2024).

**Questions:**

See weaknesses

---

> ### Author Response · Authors · 2024-11-23
> **Response to reviewer n2dh (1/2)**
>
> We thank the reviewer for their suggestions and feedback. We reply to individual comments below with references to new uploaded material in parts.
>
> > A broader set of baselines should be included in experiments, such as LERF [1], D3Field [2], and Semantic Gaussians [3].
> The results from real-world scenes (Fig. 7) are not particularly impressive; comparisons with recent methods, such as Semantic Gaussians [3] and GaussianGrasper [6], would strengthen the evaluation
>
> We agree that quantitative comparisons with such works in multi-view experiments would benefit the paper in terms of demonstrating the practical advantages of DROP-CLIP vs. Structure-from-Motion (SfM) works such as the ones mentioned by the reviewer.
> We refer the reader to Appendix A.6.3 of our updated submission for a detailed discussion and comparison with such works.
>
> > The MV-TOD dataset, while useful, is synthetic and introduces a sim-to-real gap when using modified CLIP/DINO for dense feature extraction, which limits its impact.
>
> We agree that distilling from a synthetic dataset introduces a sim-to-real domain gap that potentially influences the impact of the approach. In order to explore such an effect, we conducted the zero-shot generalization experiments in the real-world dataset OCID (Sec 4.3). As we report in our paper, our method still provides improvements over baselines trained on real-data, even though distilled from our synthetic MV-TOD.
>
> Further, we included a supplementary video (`iclr_drop_reb_supp.mp4`) where we demonstate real-time performance of DROP in real-world scenes vs. a vanilla DFF baseline based on back-projecting MaskCLIP features.
>
> > For open-vocabulary grasping, comparing the proposed 3D representation to baselines like F3RM [4], LERF-TOGO [5], and GaussianGrasper [6] would help validate its advantages.
>
> Since our approach focuses on CLIP feature distillation and not grasping, we mainly consider segmentation tasks and included our robot grasping application of Sec.4.4 as a demonstration of DROP-CLIP’s efficiency advantages over NeRF/GSplat based works (such as the ones mentioned by the reviewer), and not advantages in terms of grasping success rate
>
> In particular, as we extensively motivate in our Introduction and Related work sections, compared to Structure-from-Motion (SfM) works that rely on training NeRFs  / GSplats [1-5], our method:
>  - Does not need to collect multiple camera images, but works from single-view RGB-D. Works such as F3RM [4] make the robot grab a selfie stick and move it around in order to collect images before being able to make a representation. Other works require setups with multiple cameras fixed around the scene. DROP-CLIP assumes a single camera view mounted on the robot.
>  - NeRF / GSplat works require multiple minutes to fit the representation in the specific scene. DROP-CLIP requires 0 minutes since it has already been distilled from MV-TOD, and does not require training.
>  - NeRF works require significant time to do inference. DROP-CLIP inference is just a forward pass through the distilled encoder, so it is real-time. GSplat works with efficient implementations also provides close to real-time inference, although whether it can be real-time for feature rendering (and not just RGB rendering) is unclear.
>  - NeRF-based works are training representations that are scene-specific, and thus changing the scene (e.g. due to robot action) would require re-training the NeRF from scratch. GSplat works partially address this through deformable Gaussians, although to the best of our knowledge, deformable feature splatting has not been demonstrated in literature. DROP-CLIP works real-time, so it updates as the scene changes and doesn’t require special treatment.
>
> We hope the above clarifies the practical advantages of our method compared to the aforementioned works. We also refer the reader to our included supplementary video (`iclr_drop_reb_supp.mp4`), where we demonstrate the real-time performance of DROP in more real-world scenes.

---

> > ### Comment · Reviewer_n2dh · 2024-11-25
> >
> > Thanks for the response, but I have still some questions.
> >
> > Q1: The authors explain that "refer the reader to Appendix A.6.3 of our updated submission for a detailed discussion and comparison with such works." Although some works are included for comparisons, but the comparion is unfair, since this method takes RGBD as input while others take RGB images as input. Strongly advise the authors to include the feature distillation part of gaussiangrasper as a fair baseline that also leverages RGBD inputs.
> >
> > Q2: The authors provide a video case on real scene in supplementary video. However, the testing object is quite simple just "book" or "box", such simple categories cannot solve the concerning about sim2real gap on generalazition. Advise the authors to include some difficult cases with open category, such as "red apple", "yellow toy", which will be more convicing.

---

> > > ### Author Response · Authors · 2024-11-28
> > > **Response to comment by Reviewer n2dh**
> > >
> > > Q1. It seems that the reviewer believes that the comparison of A.6.3 is unfair due to our method using single-view RGB-D vs. 171 RGB views of baselines. The depth is required to create a sparse point-cloud which is the input to DROP-CLIP, but it's not used in baselines which establish 2D-3D correspondence via SfM. So indeed our method has an advantage for 2D<->3D mapping by having access to depth and camera parameters, instead of having to extract 2D-3D correspondence from SfM.
> > >
> > > However, remember that the task here is 2D semantic / referring segmentation, and the evaluation metrics are also computed in 2D. For SfM baselines, the metrics are computed from the rendered 2D view, while for our method from back-projecting the point-cloud to 2D. We are testing for quality of features here, depth is only used to move the features of our point-cloud back to 2D image frame so we can compute the metrics the same way as the baselines. No 3D information is actually needed to do the evaluation. Why would then depth make any difference ?  If anything, as we motivate in A.6.3, we believe this comparison is unfair towards our method, since it only uses 1 view vs. 171 of the baselines.
> > >
> > > Q2. We thank the reviewer for raising the point of complexity of tested queries. We include a visualization in our updated submission (sec A.7, Fig.20), where we compare grounding results for attribute-based queries (Coke can vs. beer can, green apple vs. red apple).

---

> ### Author Response · Authors · 2024-11-23
> **Response to reviewer n2dh (2/2)**
>
> > The novelty of the method may fall short for an ICLR submission, as it relies on projecting 2D CLIP or DINO features into 3D points via simple averaging, which is not a new approach
>
> We believe that the above comment under-represents the contributions of our work. As we extensively motivate in our Introduction and Related work sections, unlike previous works, our work does not rely on simple averaging. Instead:
>  -  it proposes a semantic informativeness metric to weight the contributions of views based on text annotations, and
>  - exploits segmentation masks to do object-wise instead of point-wise fusion.
>
> See Tab.1 for quantitative results regarding the contributions of our proposed approach compared to simple averaging.
>
> Further, the reviewer’s acknowledged novelty refers to merely the first stage of our work, as the object-centric fusion strategy is a method for improving the quality of the 3D features to-be-distilled. The second stage is actually distilling the target features with a point-cloud encoder such that during test-time, you can without training obtain object-centric 3d features from single-view RGB-D, with real-time performance and without needing segmentors to extract masks. To get there, we need a multi-view indoor cluttered dataset with broad object catalog, which is currently missing. In our work, we generate such resource and release it (MV-TOD). We hope the above clarify the contributions of our work and would appreciate suggestions for make it more clear in our main paper.
>
>
>
> [1] Kerr, Justin et al. “LERF: Language Embedded Radiance Fields.” 2023 IEEE/CVF International Conference on Computer Vision (ICCV) (2023): 19672-19682.
>
> [2] Qin, Minghan, et al. "Langsplat: 3d language gaussian splatting." Proceedings of the IEEE/CVF Conference on Computer Vision and Pattern Recognition. 2024.
>
> [3] Guo, Jun et al. “Semantic Gaussians: Open-Vocabulary Scene Understanding with 3D Gaussian Splatting.” ArXiv abs/2403.15624 (2024): n. pag.
>
> [4] Shen, Bokui (William) et al. “Distilled Feature Fields Enable Few-Shot Language-Guided Manipulation.” Conference on Robot Learning (2023).
>
> [5] Qiu, Ri-Zhao et al. “Feature Splatting: Language-Driven Physics-Based Scene Synthesis and Editing.” ArXiv abs/2404.01223 (2024): n. pag.

---

### Official Review · Reviewer_Lvif · 2024-11-02

**Soundness:** 3
**Presentation:** 3
**Contribution:** 2
**Rating:** 5
**Confidence:** 4

**Summary:**

This paper proposes a multi-view feature fusion strategy that employs object-centric priors to eliminate uninformative views based on semantic information, and fuse features at object-level via instance segmentation masks.

**Strengths:**

1. This paper addresses the issue of sub-optimal 3D feature distillation resulting from the assumption that all camera views are equally informative and presents view-independent feature distillation, allowing the extraction of 3D open-vocabulary features from a single view.
2. This paper proposes DROP-CLIP, which utilizes an object-centric prior for weighted multi-view feature re-projection, enhancing the accuracy of feature extraction.
3. A new dataset MV-TOD has been collected, which includes detailed 3D masks and 6DoF grasp pose annotations specifically designed for table-top scenes.
4. Experimental results validate the effectiveness, generalization and applicability of the proposed methods.

**Weaknesses:**

1. My main concern is the **novelty** of the proposed method. As far as the reviewer knows, the authors just replace the pixel-level features by object-level features through cropping the images. It seems that the contributions are mostly on the dataset and downstream applications, which is not enough for this venue.
2. It is beneficial to provide some comparisons with **more baselines** like sparsedff[1] or d3field[2].
3. The dataset was constructed in a simulated environment (Blender). Its **impact on real-world scenarios** is unknown. I suggest the authors provide clarification or experiments to demonstrate its generalization ability in real-world scenarios.



[1] Wang, Qianxu, et al. "Sparsedff: Sparse-view feature distillation for one-shot dexterous manipulation." *arXiv preprint arXiv:2310.16838* (2023).

[2] Wang, Yixuan, et al. "D $^ 3$ Fields: Dynamic 3D Descriptor Fields for Zero-Shot Generalizable Robotic Manipulation." *arXiv preprint arXiv:2309.16118* (2023).

**Questions:**

1. The symbols in the paper are poorly represented. I suggest the author to simplify these symbols and remove some meaningless ones.
2. Some of the fonts in the tables and figures are too small to recognize, like Tab.1, Fig.2, Fig.3.

---

> ### Author Response · Authors · 2024-11-23
> **Response to reviewer Lvif (1/2)**
>
> We thank the reviewer for their suggestions and feedback. We reply to individual comments below with references to new uploaded material in parts.
>
> > Summary: This paper proposes a multi-view feature fusion strategy that employs object-centric priors to eliminate uninformative views based on semantic information, and fuse features at object-level via instance segmentation masks.
>
> We believe that the reviewer’s summary significantly under-represents our paper’s contributions. The quoted sentence describes only our contribution regarding the multi-view feature fusion strategy (i.e. object-centric priors), which was done to improve the quality of target features to-be-distilled, but fails to acknowledge the contributions of the distillation itself. As we highlight throughout our Introduction and Conclusion sections, our work provides:
>  - MV-TOD, a synthetic dataset with dense multi-view images, cluttered tabletop scenes from 3.3k objects and mask / grasp / semantic annotations. As we motivate in Sec.2, there is no other dataset in literature that provides all such utilities in tandem. The dense multi-view asset is essential for 2D->3D feature distillation frameworks in environments other than rooms with furniture (e.g. ScanRefer, ReferIt-3D – see Sec.2) and was previously missing.
>  - After we extract 3D features with our object-centric priors (as the reviewer summarizes), we use our dataset to distill them with a point-cloud encoder that works from single-view, without needing test-time segmentors and offering real-time performance (i.e. DROP-CLIP).
>
> > My main concern is the novelty of the proposed method. As far as the reviewer knows, the authors just replace the pixel-level features by object-level features through cropping the images. It seems that the contributions are mostly on the dataset and downstream applications, which is not enough for this venue
>
> Similar to above, the reviewer just mentions 1 / 3 techniques used during generating the MV-TOD distillation targets as the sole contribution of our work (object-level 2D features through cropping). Next to that, we also: a) replace vanilla average pooling of previous works with weighted average pooling that considers semantic informativeness metric to rank view contributions and eliminate uninformative ones, and b) fuse the 2D features object-wise instead of point-wise. We refer the reviewer to Tab.1 to inspect quantitative advantages of the above contributions.
>
> Further, the reviewer’s acknowledged novelty refers to merely the first stage of our work, as the object-centric fusion strategy is a method for improving the quality of target features during train-time. The second stage is distilling the target features with a point-cloud encoder such that during test-time, you can obtain **without training** object-centric 3d features from **single-view RGB-D**, **without segmentors** and with **real-time performance**.
>
> Compare that to online feature distillation works that rely on Structure-from-Motion (SfM) datasets, obtained with NeRF / GSplats [1-5]. These works:
>  - **Need multiple camera images** to train the NeRF / GSplat (e.g. F3RM [4] makes the robot grab a selfie stick and move it around the environment to collect images).
>  - Require multiple minutes to **do training** (in order to fit the NeRF / GSplat to the specific scene) and considerable time for inference.
>  - NeRF-based methods are **not adaptable to scene changes**. Once the scene changes (e.g. due to the robot performing an action), you would have to retrain a new field. This limitation is partially addressed with deformable GSplat works, although inference time for feature splatting is unclear whether it can be real-time.
>  - **Need external segmentors** (e.g. SAM) in order to obtain the masks that will improve the quality of features with object-centric priors (LangSplat [2], F-Splatting [5], SemanticGaussians [3]).
>
> In order to make the above points more compact and also add qualitative comparison, we included a new chapter in our Appendix (A.6.3) where we extensively discuss and compare DROP with SfM approaches.
>
> > The dataset was constructed in a simulated environment (Blender). Its impact on real-world scenarios is unknown. I suggest the authors provide clarification or experiments to demonstrate its generalization ability in real-world scenarios
>
> We kindly ask the reviewer to read Sec.4.3 of our paper. In general, we agree that evaluation on real-world images is essential, and that is why we conducted the zero-shot generalization experiments in the OCID dataset, which contains real-world cluttered scenes, unseen object instances and unseen vocabulary  (see Sec.4.3 and App.A.6.1). As you can observe from the results of these sections (Tab.4, Tab.11), our work improves zero-shot semantic / referring segmentation performance compared to other zero-shot single-view approaches. Check also Appendix A.6.3 of our updated submission for further comparisons in the LERF real-world dataset.

---

> ### Author Response · Authors · 2024-11-23
> **Response to reviewer Lvif (2/2)**
>
> > It is beneficial to provide some comparisons with more baselines like sparsedff[1] or d3field[2].
>
> We thank the reviewer for bringing up such works, which are conceptually close to our paper, since they do not rely on NeRFs / GSplats to do feature distillation, but use depth to project the 2D feature and then improve consistency with point-pruning (SparseDFF). Such works maintain the real-time requirement that our work also strives for, although still having several differences. In particular:
>  - Neither SparseDFF nor D3Field use CLIP features and evaluate for semantic / referring segmentation. SparseDFF uses DINOv2 features to obtain pose descriptors for imitation learning. D3Fields directly distills SAM masks, so there are no CLIP feature to do similarity computation with text queries, but just instance segmentation.
>  - Both works rely on sparse reconstruction, i.e. needing 4 views to work, compared to 1 of DROP.
>
> Adjusting these works to our setup (MaskCLIP features instead of SAM / DINO for referring segmentation tasks and single-view reconstruction), then they become equivalent to our  MaskCLIP->3D baseline used in our paper, albeit the point-pruning step (introduced in SparseDFF).
>
>  We include a new supplementary video where we illustrate advantages of DROP vs. vanilla DFFs in terms of feature consistency and grounding performance, incl. real-world scenes (see `iclr_drop_reb_supp.mp4`).
>
> > The symbols in the paper are poorly represented. I suggest the author to simplify these symbols and remove some meaningless ones.
>
> We used standard mathematical notation used in literature (e.g. OpenScene [6], OpenMask3D [7]). We would appreciate some further feedback regarding which symbols are poorly represented / meaningless, so we can improve our methodology section.
>
> > Some of the fonts in the tables and figures are too small to recognize, like Tab.1, Fig.2, Fig.3.
>
> In our new submission, we increased the font of text in Fig.2 and Fig.3 and increased the overall size of Tab.1 to address the reviewer’s point. We kindly ask the reviewer to let us know whether the fonts are comprehensive in this draft (can also use zoom in the pdf viewer).
>
> [1] Kerr, Justin et al. “LERF: Language Embedded Radiance Fields.” 2023 IEEE/CVF International Conference on Computer Vision (ICCV) (2023): 19672-19682.
>
> [2] Qin, Minghan, et al. "Langsplat: 3d language gaussian splatting." Proceedings of the IEEE/CVF Conference on Computer Vision and Pattern Recognition. 2024.
>
> [3] Guo, Jun et al. “Semantic Gaussians: Open-Vocabulary Scene Understanding with 3D Gaussian Splatting.” ArXiv abs/2403.15624 (2024): n. pag.
>
> [4] Shen, Bokui (William) et al. “Distilled Feature Fields Enable Few-Shot Language-Guided Manipulation.” Conference on Robot Learning (2023).
>
> [5] Qiu, Ri-Zhao et al. “Feature Splatting: Language-Driven Physics-Based Scene Synthesis and Editing.” ArXiv abs/2404.01223 (2024): n. pag.
>
> [6] Peng, Songyou et al. “OpenScene: 3D Scene Understanding with Open Vocabularies.” 2023 IEEE/CVF Conference on Computer Vision and Pattern Recognition (CVPR) (2022): 815-824.
>
> [7] Takmaz, Ayca et al. “OpenMask3D: Open-Vocabulary 3D Instance Segmentation.” ArXiv abs/2306.13631 (2023): n. pag.

---

> > ### Comment · Reviewer_Lvif · 2024-12-02
> > **Official Comment by Reviewer Lvif**
> >
> > I thank the authors for the clarifications.
> >
> > Most of my concerns are addressed, and I think this is essentially a good work. However, I am still concerned about  the contribution and I think this is also a big cencern for other reviewers. Thus I suggest the author clearly list the unique theoretical contribution in the manuscript to help the reviewers to identify.
> >
> > Therefore, I will increase my score to 5.

---

### Official Review · Reviewer_FXeK · 2024-11-04

**Soundness:** 2
**Presentation:** 2
**Contribution:** 2
**Rating:** 5
**Confidence:** 4

**Summary:**

This paper proposes a method for fusing 2D semantic features from foundation models into 3D representations, applying this approach to downstream tasks in robotics, such as manipulation. The focus is on leveraging information across different views to maximize useful insights, while also addressing the fusion of priors in an object-centric manner. Additionally, the paper introduces a new dataset designed to tackle the challenges posed by cluttered scenes in real-world environments.

**Strengths:**

Two areas that are less well addressed in the literature include:

Semantics-informed View Selection: This approach utilizes an informative matrix to balance information across different views while accounting for uncertainty.

Object-wise Fusion and Features: The emphasis on object-centric features is potential to enhance generalization in complex scenes.

Table 1 provides an informative summary of the datasets.

**Weaknesses:**

My primary concern lies in the fact that feature distillation has been a popular topic in recent research, making it difficult to identify the paper's key contributions in this context. While the dataset is certainly valuable for the community, the fundamental challenge of achieving multi-view consistency when fusing 2D features into 3D representations remains a critical issue that the paper does not fully address.

The robotics experiments presented are interesting; however, they do not effectively demonstrate the performance of the proposed method. The research presented in this paper only finds the parts of interested objects. As noted in the discussion, the current manipulation pipeline appears to be limited to a two-stage open-loop demonstration.

**Questions:**

How can we ensure that the fused 3D features are consistent across multiple views?

---

> ### Author Response · Authors · 2024-11-23
> **Response to reviewer FXeK**
>
> We thank the reviewer for their suggestions and feedback. We reply to individual comments below with references to new uploaded material in parts.
>
> > My primary concern lies [...] paper's key contributions in this context.
>
> We have tried to extensively highlight the contributions of our work compared to present works throughout the Introduction and Related Work section. We briefly summarize here for the consideration of the reviewer.
>
> Compared to online feature distillation works that rely on Structure-from-Motion (SfM) datasets, obtained with NeRF / GSplats [1-5] these works:
>  - **Need multiple camera images** to train the NeRF / GSplat (e.g. F3RM [4] makes the robot grab a selfie stick and move it around the environment to collect images).
>  - Require multiple minutes to **do training** (in order to fit the NeRF / GSplat to the specific scene) and considerable time for inference.#
>  - NeRF-based methods are **not adaptable to scene changes**. Once the scene changes (e.g. due to the robot performing an action), you would have to retrain a new field. This limitation is partially addressed with deformable GSplat works, although inference time for feature splatting is not yet real-time.
>  - **Need external segmentors** (e.g. SAM) in order to obtain the masks that will improve the quality of features with object-centric priors (LangSplat [2], F-Splatting [5], SemanticGaussians [3]).
>
> In contrast, DROP-CLIP doesn’t need test-time segmentors, works from a single RGB-D image and runs real-time (so can also capture scene changes without special treatment). We also highlight the above points during Sec.3.3 of our paper. We would appreciate further feedback by the reviewer regarding how to better clarify the above point in our paper.
>
> Compared to offline feature distillation works that, similar to DROP-CLIP, use point-cloud encoders to distill multi-view fused features from a dataset (e.g. OpenScene [6] ), our work:
>  - generates and trains in MV-TOD, a multi-view cluttered tabletop scene dataset, instead of room scan datasets, which contains dense multi-view images, broad variety of objects and semantic concept annotations (see Sec.2 and App.A for details).
>  - introduces object-centric priors to improve the quality of the target features that will be distilled (see Sec.4.1 for qualitative comparisons with previous works).
>  - demonstrates with ablations (Sec.4.1), in-distribution (Sec.4.2) and out-of-distribution (Sec.4.3) experiments that the distilled features significantly improve performance for single-view semantic / referring segmentation tasks compared to previous works.
>
> We also refer the reader to Appendix A.6.3 of our updated submission for extended discussion and comparison with modern NeRF / GSplat approaches in a scene of the LERF dataset
>
> > How can we ensure that the fused 3D features are consistent across multiple views?
>
> If the reviewer refers to the fused 3D feature during train time (the result of the object-centric feature fusion) we can ensure that the features are view-independent since we explicitly construct them by fusing across views.
>
> If the reviewer refers to the reconstructed features from DROP-CLIP at test-time, we agree that our approach does not guarantee that the features will be consistent across views, but it encourages it during training. In particular, by training on a dataset such as MV-TOD which contains 73 views for each scene, and forcing the same target features for each one of the views during training, we encourage DROP to produce the same features regardless of the input view.
>
> From an architectural perspective, since the input to our encoder is a point-cloud that has been transformed to world frame and mean-shifted (using the camera extrinsics), the method can guarantee translation-invariance (so camera position). Regarding rotation-invariance, it is true that our MinkUNet architecture we're using cannot guarantee it. However, we apply extensive rotation augmentations to the 73 scene views during training to make the network robust to camera orientation.
>
> To demonstrate view independence qualitatively, we include a supplementary video where we move the camera or the object-of-interest in real-time and observe the changes in the feature space between CLIP and a standard DFF that relies on vanilla MaskCLIP back-projection (see `iclr_drop_reb_supp.mp4`).
>
> >The robotics experiments [...] a two-stage open-loop demonstration
>
> The focus in our paper was on distilling CLIP features and therefore we wish to mostly demonstrate advancements in 3D segmentation tasks, and not robotic grasping. We do not claim our robotic pipeline is optimal in terms of grasping success rates, and was added as a demonstration of the practical advantages of our work compared to modern NeRF / GSplat works for CLIP feature distillation (as mentioned above and discussed in Appendix A.6.3 of our updated pdf, no need for multiple views, real-time, immune to scene changes, no need for segmentors at test-time).

---

> > ### Author Response · Authors · 2024-11-23
> > **Response to reviewer FXeK [references]**
> >
> > [1] Kerr, Justin et al. “LERF: Language Embedded Radiance Fields.” 2023 IEEE/CVF International Conference on Computer Vision (ICCV) (2023): 19672-19682.
> >
> > [2] Qin, Minghan, et al. "Langsplat: 3d language gaussian splatting." Proceedings of the IEEE/CVF Conference on Computer Vision and Pattern Recognition. 2024.
> >
> > [3] Guo, Jun et al. “Semantic Gaussians: Open-Vocabulary Scene Understanding with 3D Gaussian Splatting.” ArXiv abs/2403.15624 (2024): n. pag.
> >
> > [4] Shen, Bokui (William) et al. “Distilled Feature Fields Enable Few-Shot Language-Guided Manipulation.” Conference on Robot Learning (2023).
> >
> > [5] Qiu, Ri-Zhao et al. “Feature Splatting: Language-Driven Physics-Based Scene Synthesis and Editing.” ArXiv abs/2404.01223 (2024): n. pag.
> >
> > [6] Peng, Songyou et al. “OpenScene: 3D Scene Understanding with Open Vocabularies.” 2023 IEEE/CVF Conference on Computer Vision and Pattern Recognition (CVPR) (2022): 815-824.

---

> > ### Comment · Reviewer_FXeK · 2024-11-25
> > **Discussion**
> >
> > I appreciate the detailed responses provided by the authors, which have helped clarify the motivation behind training a single-view 3D semantic feature extractor and its value. I have reviewed the responses to other reviewers as well, but I still have strong concerns, which align with those raised by Reviewer Lvif and Reviewer n2dh.
> >
> > **Novelty and Theoretical Contribution to ICLR**
> >
> > I appreciate the motivation of introducing object-centric prior as a valuable part of generalization over multiple scenes in the real world. However, technically, the object-prior in the paper is achieved solely by segmenting object instances via masks and fuse features independently. I was expecting more insightful prior distribution, such as pre-trained large text-to-3D models, that can be used to learn diverse, multi-categories, and complicated object priors.
> >
> > As for the response to Reviewer Lvif regarding the changes made by (1) replacing pooling and (2) fusing at the object level, I am not convinced that these changes represent a significant technical and theoretical advancement.
> >
> > **Effectiveness/Generalization, Especially in the Real World**
> >
> > Several reviewers have highlighted the limitations of real-world experiments. While I appreciate the addition of a single sequence experiment in the Appendix, I still have fundamental concerns regarding the generalization ability of the point cloud encoder to produce robust 3D features, considering that fundamentally, this method is trained on synthetic datasets.
> >
> > **Conclusion**
> >
> > Overall, I feel the theoretical contribution may not be substantial enough for an ICLR-level venue. I would recommend considering a robotics conference like ICRA, which is more application-oriented.

---

> > > ### Author Response · Authors · 2024-11-28
> > > **Comment on discussion raised by Reviewer FXeK**
> > >
> > > We thank the reviewer for discussing the overall topic of novelty and contributions. We would like to comment on some of the points raised here.
> > >
> > > > However, technically, the object-prior in the paper is achieved solely by segmenting object instances via masks and fuse features independently. I was expecting more insightful prior distribution, such as pre-trained large text-to-3D models, that can be used to learn diverse, multi-categories, and complicated object priors.
> > >
> > > Indeed, instance masks and object-wise fusion are the spatial priors introduced by our method. However, we also included a semantic prior, which is implemented via the semantic informativeness metric. This metric uses text data (describing objects at instance-level, so attributes, colors, materials etc. -- more than multi-categories) to refine the 3D features. These text data are generated from large VLMs (as we describe in Sec.2). We believe than even though not using text-to-3D models and object-part annotations, which the reviewer refers to, our text data generation and inclusion in object priors is a clear step towards this direction, not demonstrated in previous works.
> > >
> > > > Several reviewers have highlighted the limitations of real-world experiments. While I appreciate the addition of a single sequence experiment in the Appendix, I still have fundamental concerns regarding the generalization ability of the point cloud encoder to produce robust 3D features, considering that fundamentally, this method is trained on synthetic dataset
> > >
> > > As we explicitly highlight in our limitation section, indeed, our encoder cannot guarantee robust 3D features in any in-the-wild real-world scenario, as it is dependent on the object catalog included in our training dataset. However, in our work we show for objects that are within the training distribution (in terms of both geometry and semantics), our method provides improvements over pre-trained methods, even in real-world (Sec.4.3,4.4), even though its trained on synthetic data.
> > >
> > > We do not claim to have a 3D foundation model that works for all scenarios or anything like that. Instead, we propose a new methodology for reconstructing object-centric 3D CLIP features from 2D, based on offline distillation in tabletop-specific dataset, and show that it works better than previous 2D approaches in typical household objects, as proof of concept. We believe that If you scale the training dataset MV-TOD (e.g. via using text-to-3D models to scale up the object catalog, as we mention in line 593) , you will get the desired generalization.

---

### Official Review · Reviewer_6oxN · 2024-11-04

**Soundness:** 3
**Presentation:** 3
**Contribution:** 3
**Rating:** 6
**Confidence:** 4

**Summary:**

The paper proposes a novel method for 2D->3D feature distillation using point cloud encoder, but focus on the multi-view fusion strategy. In particular, the paper is centered around the distillation of pretrained 2D CLIP models into 3D encoder, dubbed DROP-CLIP. Compared to existing multi-view fusion and per-scene optimization method, the proposed method does not require expensive optimization, and work with as few as a single RGB-D view. To facilitate the training of DROP-CLIP, the paper also uses blender to construct a large synthetic dataset termed MV-TOD for training the 3D encoder. Experiments show significant improvement of segmentation results on the MV-TOD dataset over existing baselines (OpenScene and OpenMask3D).

**Strengths:**

- **Technical contributions.** The paper proposes a series of techniques and datasets that are well-justified. To start with, viewpoint uncertainty is an important factor in 2D->3D distillation. Many existing methods suffer from inaccurate predictions caused by such uncertainty. The paper not only proposes a method to address this, but proposes an object-centric dataset that attempts to eliminate the bias of room-scale datasets. The individual components are also well-ablated in Tab. 2.
- **Good ablation and motivating applications.** The experiment sections ablate individual components well and show improvement over existing baselines on the proposed MV-TOD dataset. Though more real-world data experiments would be appreciated, the experiments seem to justify each component to some extent.

**Weaknesses:**

- **Unclear performance on real-world data.** Though the paper demonstrates that DROP-CLIP outperforms recent methods (OpenScene and OpenMask3D), the results were obtained on the synthetic MV-TOD dataset. For the experiments with real-world data in Tab. 4, the paper compares with relatively outdated methods. So the improvement of DROP-CLIP on real-world data seems a bit inconclusive.
- **Efficiency is unreported.** The paper provides a motivating application that uses DROP-CLIP for robotics grasping, which is good. However, efficiency is crucial for robotics applications. The paper is unclear as to whether the proposed method is real-time.
- **Unclear pipeline.** The paper does not seem to have a figure that shows how all components are connected. For example, L240-L242 mentions that sets of 2D segmentation masks are used. However, the method overview figure does not show how 2D masks are integrated, and there does not seem to be a description of how these masks are obtained (especially for real data) in the main paper.
- **Justification of the dataset contribution.** In the introduction, the paper discusses the necessity of constructing the MV-TOD dataset from synthetic data by comparing it to an existing room-level dataset. However, there is no quantitative evidence to support this. For example, how would the proposed component work if it is trained on an existing dataset and evaluated on the proposed dataset? Would joint-training improve the overall performance?
- **Discussion of related work.** The paper compares with several 2D->3D distillation methods. However, there are several more recent methods that are closely related to the paper. Specifically, [A] and [B] also distill 3D features, and [B] also uses 2D input masks to improve 3D segmentation results. Citing and providing differentiation with these work would benefit the presentation.

[A] Qin, Minghan, et al. "Langsplat: 3d language gaussian splatting." Proceedings of the IEEE/CVF Conference on Computer Vision and Pattern Recognition. 2024.

[B] Qiu, Ri-Zhao, et al. "Feature Splatting: Language-Driven Physics-Based Scene Synthesis and Editing." ECCV. 2024.

**Questions:**

Reflecting on weaknesses above, my questions are

- Is it possible to include additional experiments on real-world data?
- What is the efficiency of the proposed method?
- Can the figures be improved to be more informative?
- Is it possible to do joint training on both MV-TOD and some other datasets to show possible improvement? If not, why?
- Discussion of related work.

---

> ### Author Response · Authors · 2024-11-23
> **Response to Reviewer 6oxN**
>
> We thank the reviewer for their suggestions and feedback. We reply to individual comments below with references to new uploaded material in parts.
>
> **Unclear performance in real-world data** Unfortunately the OCID dataset, for which we conducted real-world evaluations in Tab.4, provides only single-view images, so there is no means of comparison with approaches that require multiple views. That is why we only conducted comparisons with 2D CLIP-based approaches LSeg, OpenSeg and MaskCLIP, which are the de-facto choices for 2D zero-shot semantic segmentation.
>
> However, in order to address the reviewer’s concern, we also conducted comparative experiments with modern Structure-from-Motion (SfM) methods LERF [1], LangSplat [2] and SemanticGaussians [3] in the localization and semantic segmentation tasks in a scene of the LERF dataset. We refer the reader to the A.6.3 chapter of our Appendix in the updated submission.
>
> **Unreported efficiency** We mention throughout the paper (lines 67, 95, 299, 520) that DROP-CLIP achieves real-time performance, since inference is a simple forward pass from the distilled encoder. In particular, we can achieve ~25 fps on a RTX-3090 GPU for obtaining the feature-clouds illustrated in our paper (e.g. Fig.8, Fig.18), without considering PCA visualizations.
>
> Further, we include more qualitative results, where we wish to show the real-time efficiency and view-independence of CLIP vs. a vanilla DFF baseline that relies on MaskCLIP feature back-projection (see included supplementary video `iclr_drop_reb_supp.mp4`).
>
> **Unclear pipeline** The segmentation masks are provided as part of the training dataset MV-TOD, which generates them automatically from the Blender engine. We highlight however that segmentation masks are needed only at train-time, as a means to improve the quality of the fused 3D features that will be distilled. Masks are not needed at test-time. As we mention in Sec.3,3:
> "*Importantly,no labels,prompts,or segmentation masks are needed at test-time to reproduce the fused feature-cloud [...]*~
>
> So even in real-data, you don’t need segmentation masks. DROP-CLIP has learned to reproduce features that were constructed with the aid of segmentation masks, but doesn’t need segmentation masks itself. We include this point in the updated submission (line 241)  to address the reviewer’s point.
>
> **Justification of the dataset contribution** The main limitation of previous room scan data, as we motivate in Sec.2, is that:
>   - They only contain category-level annotations from a small fixed set (e.g. 51k expressions for 18 unique objects in ScanRefer), while MV-TOD has open-vocabulary language annotations generated with VLMs (671k expressions for 149 categories).
>  - They do not contain cluttered tabletop scenarios, but mostly furniture objects in room layouts. MV-TOD contains cluttered tabletop scenes from over 3300 unique object CAD models.
> We refer the reader to Tab.1 for a more comprehensive comparison with existing datasets.
>
> > Is it possible to do joint training on both MV-TOD and some other datasets to show possible improvement? If not, why?
>
> The focus of this work is on robotic manipulation applications, where we mostly care about household-like objects that could appear in tabletops. Previous datasets such as ScanRefer or ReferIt-3D focus on room layouts, so objects are mostly furniture (chairs, sofas, counters etc.). It is considerable that one could filter out scenes from such datasets that feature tabletops and clutter in order to complement the MV-TOD dataset. However, just from a pure numbers perspective, we believe the contribution of such datasets would be minimal considering the extremely narrow distribution of household objects that appear.
>
> **Discussion of related work.** We thank the reviewer for providing additional references. We already had a comparison with LangSplat [2] in our extended related work section (App.A-8), which we re-write in our updated submission, in order to include more related works, including the ones mentioned by the reviewer.
>
> [1] Kerr, Justin et al. “LERF: Language Embedded Radiance Fields.” 2023 IEEE/CVF International Conference on Computer Vision (ICCV) (2023): 19672-19682.
>
> [2] Qin, Minghan, et al. "Langsplat: 3d language gaussian splatting." Proceedings of the IEEE/CVF Conference on Computer Vision and Pattern Recognition. 2024.
>
> [3] Guo, Jun et al. “Semantic Gaussians: Open-Vocabulary Scene Understanding with 3D Gaussian Splatting.” ArXiv abs/2403.15624 (2024): n. pag.

---

> > ### Comment · Reviewer_6oxN · 2024-11-23
> > **Feedback**
> >
> > I thank the authors for the clarifications.
> >
> > Though most of my concerns are addressed, my main concern is the lack of evidence on real data. I do think that there are some more recent 2D zero-shot segmentation works that can be compared with (e.g., GroundedSAM), which would help highlight the contributions of the work.
> >
> > In summary, I maintain a score of 6.

---

> > > ### Author Response · Authors · 2024-11-28
> > > **Inclusion of GroundedSAM**
> > >
> > > We thank the revieiwer for their comment. We updated our pdf submission with results with GroundedSAM in the semantic segmentation task of OCID in Sec.4.3, Table 4. DROP-CLIP surpasses the GroundedSAM scores by 2.7% in the mAcc@75 metric. We find that the GroundingDINO part of GroundedSAM mostly fails to produce a good bounding box for OCID-VLG queries. Note that unlike CLIP-based methods that compute similarities, no negative prompts are provided for GroundedSAM.

---

### Meta-Review · Area_Chair_GqXH · 2024-12-23

**Metareview:**

**Summary**

The paper aims to improve distillation of semantic information from 2D features (e.g. CLIP) into 3D by proposing to fuse object-level features from multiple views by taking into account which views are more informative.

To study this, MV-TOD, a synthetic dataset of multi-view images of cluttered tabletops is created.  The dataset consists of 15K tabletop scenes with dense multi-view renderings (73 cameras per scene), and detailed object information (provided by vision-language model) for each object.

Experiments are conducted on this dataset comparing the proposed approach (DROP-CLIP) against prior work on different segmentation tasks (semantic, instance, language-based query).  Ablations are also conducted to compare fusing at different levels, and the effect of different number of views.  The method is also show to generalize to real-world scenes and can be used in language-guided robotics grasping.

**Strengths**

Reviewers noted the following strengths of the work:
1. Distilling 2D features from vision-language models into 3D is a timely topic with lots of activity. [FXeK]
2. The proposed method (DROP-CLIP) looks at two under-studied aspects of the problem [FXeK]:
   a. Semantic informed view selection
   b. Object-centric fusion of features
3. Generated dataset (MV-TOD) of cluttered tabletops can be valuable for the community [FXeK,6oxN,Lvif]
4. Good ablations [6oxN,n2dh] and experimental results validate the approach [Lvif]

The AC believe the dataset and the proposed method can provide value to the community.

**Weaknesses**

Reviewers noted the following weaknesses:
1. Positioning with prior work need to be improved
   - Contributions relative to other recent works in 2D to 3D feature distillation is unclear [FXeK]
   - Important related work is missed [6oxN]
   - Novelty and contribution wrt to prior work [n2dh,FXeK,Lvif]
2. Missing important baselines [6oxN,n2dh,Lvif]
   - Reviewers noted that the methods compared against in the main paper are somewhat outdated [6oxN] with suggestions of other methods to compare against [6oxN,n2dh,Lvif]
3. Some parts of the method are unclear [6oxN]
4. Doubts about performance on target application of robotic manipulation (which is the focus of this work).
   - No discussion of efficiency for real-time robotics [6oxN]
   - Manipulation pipeline is limited [FXeK]
5. Concerns about generalization and performance on real-world data [6oxN, Lvif]
   - Results on real-world scenes are not so impressive [n2dh]
6. Usefulness of MV-TOD (synthetic dataset) [n2dh]

While some of the above where addressed during the author response period, reviewers still had concerns regarding the performance of the proposed method on real-world data and whether the proposed method will work effectively for real-world robotics manipulation.  In addition, reviewers felt the technical contributions of the work was sufficient for ICLR.

**Recommendation**

Overall, reviewers were borderline negative on this work, questioning the contribution of this work and how well the method works on real-world data.
The AC believes that there is value in this work, but in its current state, it is not ready for acceptance at ICLR 2025.  As the work is more application focused, reviewers recommend a robotics venue such as ICRA.

**Additional Comments On Reviewer Discussion:**

The paper received marginally negative ratings (with 3 reviewers being slightly negative [n2dh,Lvif,FXeK] with a score of 5 and one reviewer being slightly positive [6oxN] with a score of 6).  Initially reviewer Lvif was more negative (starting score of 3), but some of their concerns were addressed during the author response period and they raised their score to 5.

The biggest concerns from the reviewers were positioning with respect to prior work including missing discussion / comparison with recent work, and questions about whether the proposed method would work well on real-world robotic manipulation tasks.  Improvements to the manuscript and additional experiments provided by the authors during the author response period helped to alleviate some of the concerns of reviewers.  However, reviewers still are not positive on the work stating concerns about how well the model would work on challenging real-world data with complex language.

In addition, several reviewers [n2dh,FXeK,Lvif] expressed the view that they felt the novelty and theoretical contribution to ICLR is not sufficient.  The AC does not believe that to be a serious concern, as small insights can be important and empirical findings are also valuable contributions.  Reviewers also point out that there has been many recent work that aims to do distillation of information from 2D to 3D.  It is not clear to the AC whether after the author response period, there is sufficient discussion and comparison against recent work, and whether the experiments and insights provided by the work is sufficient value to the community to warrant acceptance.

---

### Decision · Program_Chairs · 2025-01-22

Reject